



# Tropospheric eddy feedback to different stratospheric conditions in idealised baroclinic life cycles

Philip Rupp[1] and Thomas Birner[1]

[1]Meteorological Institute Munich, Ludwig-Maximilians-University, Munich, Germany

**Correspondence:** Philip Rupp (philip.rupp@lmu.de)

**Abstract.**

A pronounced signature of stratosphere-troposphere coupling is a robust negative anomaly in the surface northern annular mode (NAM) following major sudden stratospheric warming (SSW) events, consistent with an equatorward shift of the tropospheric jet. It has previously been pointed out that tropospheric eddy feedbacks, mainly induced by anomalies in the lowermost extratropical stratosphere, play an important role in creating this surface NAM-signal. We use the basic setup of idealised baroclinic life cycles to investigate the influence of stratospheric conditions on the behaviour of tropospheric synoptic-scale eddies. Particular focus is hereby given on the enhancement of the tropospheric eddy response by surface friction, as well as the sensitivity to wind anomalies in the lower stratosphere. We find systems that include a tropospheric jet only (modelling post-SSW conditions) to be characterised by an equatorward shift of the tropospheric jet in the final state of the life cycle, relative to systems that include a representation of the polar vortex (mimicking more undisturbed winter-time conditions), consistent with the observed NAM-response after SSWs. The corresponding surface NAM-signal is increased if the system includes surface friction, presumably associated with a direct coupling of the eddy field at tropopause level to the surface winds. We further show that the jet shift signal observed in our experiments is mainly caused by changes in the zonal wind structure of the lowermost stratosphere, while changes in the wind structure of the middle and upper stratosphere have almost no influence.

## 1 Introduction

### 1.1 General background

Troposphere and stratosphere form a dynamically coupled system. In order to better understand tropospheric weather and climate behaviour it is essential to understand how stratospheric conditions and processes can have a downward influence and modify the tropospheric circulation or produce surface signals.

The maybe most prominent stratospheric phenomena in the northern hemisphere are (major) sudden stratospheric warming (SSW) events. During these sudden warmings the westerly winds of the stratospheric polar jet (also polar vortex) break down or even reverse. Thompson and Wallace (1998) showed that the winter time variability of the stratospheric polar vortex and the tropospheric mid-latitude jet are strongly correlated.





Baldwin and Dunkerton (2001) used a composite study of weak vortex events (of which SSWs would form the extreme

subset) to investigate the time dependence of this coupling in more detail. They showed how the stratospheric zonal wind anomalies propagate downwards into the troposphere and demonstrated that this downward influence appears to have two components: at first the signal modifies the wind structure above and at tropopause level on sub-seasonal to seasonal time scales, creating a long-lasting zonal wind anomaly in the lower stratosphere. From there the signal can penetrate quasi-instantaneously into the troposphere and create surfaces anomalies that persist on weekly time scales.

Since then various studies have supported the idea that the break-down of the polar vortex can have a downward impact and induce zonal wind anomalies in the troposphere. In particular, it can lead to periods with weak and equatorward shifted tropospheric jet stream. This equatorward shift of the jet typically manifests as negative anomaly of the northern annular mode (NAM) index or similar indices (e.g., Karpechko et al., 2017; Charlton-Perez et al., 2018). Changes in the large scale tropospheric circulation can then affect local surface weather and, with it, change the likelihood of extreme events like cold

spells (Thompson and Wallace, 2001; Kolstad et al., 2010; Kautz et al., 2020).

Different mechanisms have been proposed to explain the downward propagation of stratospheric wind anomalies and their influence on the tropospheric circulation. However, no single fully conclusive mechanism has been found yet. Note that, in addition, the tropospheric response to SSWs might also be caused by a combination of different coupling processes. One of these potential coupling processes is given by tropospheric eddy feedback as a response to the induced stratospheric anomalies.

Domeisen et al. (2013) have shown in idealised model runs that tropospheric eddies are essential to obtain a robust negative NAM signal following a SSW. Hitchcock and Simpson (2014) also found tropospheric eddy feedback to play a significant role in creating a NAM-like surface response. They further concluded that the most relevant aspect of the stratospheric variability does not seem to be the wind reversal in the mid-stratosphere, but the persistent wind anomalies in the lowermost stratosphere. Karpechko et al. (2017) showed that in both, model runs and reanalysis data, SSWs which produce strong and long-lasting

anomalies in the lowermost stratosphere have an increased likelihood for a tropospheric impact compared to SSWs with weak anomalies in the lowermost stratosphere.

### 1.2   Previous baroclinic life cycle work relevant for this study

A simple, yet fundamental, way to investigate the role of synoptic scale eddies in the dynamical coupling between stratosphere and troposphere is through (idealised) baroclinic life cycle experiments, an initial value problem starting from an imposed baro-

clinically unstable tropospheric jet. During the subsequent break-down of the imposed jet a baroclinic wave can be observed to develop, grow and eventually decay, leaving the system in a state with a more barotropic, strengthened and poleward shifted jet compared to the initial conditions (see, e.g., Simmons and Hoskins, 1978; Thorncroft et al., 1993). Such life cycle experiments have previously been used to study the influence of stratospheric winds onto the evolution of tropospheric baroclinic eddies.

Wittman et al. (2004) performed idealised life cycle experiments using initial conditions that either do or do not include

winds in the stratosphere, representing situations with an intact or a broken-down polar vortex. They found that if the system includes a polar vortex the evolution of the life cycle is strongly modified and when the polar vortex is removed the system exhibits a (weak) dipole structure in the surface geopotential height field, similar to the surface NAM response observed after





SSWs, which corresponds to an equatorward shift of the tropospheric jet. They further note that this surface signal is weak
if the polar vortex is rather confined to the stratosphere, but gets strongly enhanced if the polar vortex reaches deep into the
troposphere.

In a following study Wittman et al. (2007) investigated the role of stratospheric vertical shear onto the evolution of baroclinic
life cycles. They used three different setups in which the winds of the tropospheric jet either decreased, stayed constant or
(further) increased above the jet core. For the three situations they found pronounced differences in the evolution of the life
cycle, including substantial changes in the growth rate of the baroclinic waves and the qualitative characteristics of the wave
growth and decay phases. It should be noted, that the initial conditions used by Wittman et al. (2007) were mostly motivated to
resemble a setup of the Eady model for baroclinic instability, rather than realistic atmospheric conditions. The corresponding
change of stratospheric shear induces strong changes in the vertical curvature of zonal wind at tropopause level, and thus strong
changes in the meridional gradient of potential vorticity (PV) in that region.

Kunz et al. (2009) used a similar setup as Wittman et al. (2004) and also found that the presence of a stratospheric jet can
qualitatively alter the evolution of the baroclinic life cycle. Further, they could not explain the modification of the life cycle with
simple refractive index linear theory and therefore concluded that the non-linear part of the wave evolution plays an important
role in the coupling.

Smy and Scott (2009) investigated the influence of stratospheric PV anomalies on the evolution of idealised baroclinic life
cycles to obtain insights into the dynamical coupling of stratosphere and troposphere during and after SSWs. They reported
a decrease in growth rates and general wave activity (and a corresponding reduction in magnitude of the surface geopotential
anomaly of the final state) with increasing perturbation amplitude. Smy and Scott (2009) further comment on the influence of
the 'sub-vortex region', given by the lowermost extratropical stratosphere. A modification of the wind structure (or equivalently
the PV field) in this region can represent the direct effect of stratospheric anomalies on the tropospheric winds. They found that
the influence of the polar vortex on the life cycle evolution decreases as the stratospheric jet reaches deeper into the lowermost
stratosphere. These results partly seem to be in disagreement with the results of Wittman et al. (2007) or Kunz et al. (2009).
However, Smy and Scott (2009) also note that their results might be explained by a change in tropospheric horizontal shear
due to the non-local effects of the stratospheric PV anomaly and a corresponding fundamental change in the nature of the life
cycle (see also Thorncroft et al. (1993) for details on how horizontal shear can affect the evolution of baroclinic waves).

While much focus was given on sensitivities of the linear growth phase of baroclinic life cycles to various changes of
the system Barnes and Young (1992) also investigated the evolution during the non-linear decay phase to a range of flow-
dependent forcing processes, including surface friction. They found the system to undergo a series of growth and decay phases
in cases with sufficiently weak diffusion, in contrast to the single growth phase with subsequent decay of eddy energy in
cases with strong diffusion. They further showed that simulations with surface friction can produce more pronounced such
'secondary cycles', i.e., growth and decay phases following the initial life cycle, as the surface drag tends to work against the
barotropisation of the non-linear phase and thus act as source of baroclinicity.





### 1.3  Potential influence of surface friction

The influence of surface friction onto the evolution of baroclinic eddies is potentially crucial to understand the surface signal observed after SSWs, as it can be argued that the inclusion of surface friction increases the potential for the mid- and upper-tropospheric eddy field to couple to the surface winds. This can be illustrated using the evolution equation of the vertically averaged zonal mean zonal wind, given in Equation 1 (see, e.g., chapter 10 of Vallis (2017)).

$$\partial_t [\bar{u}] = -\partial_y \left[ \overline{u'v'} \right] - \bar{u}_{sfc}/\tau, \tag{1}$$

where $u$ and $v$ are zonal and meridional wind, $\bar{u}_{sfc}$ the zonal mean zonal surface wind, $\tau$ the surface friction time scale, square brackets and overbars denote vertical and zonal averages, respectively, and primed quantities describe deviations from the zonal mean (note that we neglected the mean flux term as it tends to be small in our system, consistent with quasi-geostrophic scaling). Here we used a linear damping of surface winds as simple parametrisation of surface friction. In the case with vanishing friction ($\tau \to \infty$), only the meridional momentum fluxes can act as source for (vertically averaged) zonal momentum and changes in $\bar{u}$ tend to occur in regions of non-zero momentum flux, i.e., around tropopause level for baroclinic life cycle experiments. For finite values of $\tau$, on the other hand, the atmosphere can 'exchange' momentum with the surface, allowing for a non-local coupling between surface winds and the eddy field. This additional coupling mechanism suggests that a dynamic modification of the eddy field (due to the presence of a stratospheric jet) can lead to an enhanced change of the corresponding surface winds (in terms of the difference between final and initial state) in cases where surface friction is active in the system.

### 1.4  Structure of this study

In the present paper we further investigate what impact the presence of a stratospheric polar vortex has on the idealised tropospheric baroclinic life cycle. In particular we are interested in the sensitivity of the life cycle evolution to changes in wind structure in the lower stratosphere, compared to changes in the middle and upper stratosphere, and the influence of surface friction onto the surface signal of the life cycle induced by the presence of a stratospheric jet. We hereby mostly focus on the modification of the equilibrated 'final' state of the system, as opposed to the details of the (linear) growth stage or the (non-linear) decay stage of the baroclinic wave.

Section 2 introduces the model setup used in this study and lays out the specifics of the different sets of initial conditions. In Section 3 we discuss in detail various changes of the evolution of the baroclinic life cycle due to the presence of stratospheric jet, with particular focus on the NAM-like response of the troposphere in the final state of a life cycle when there is no stratospheric jet present, compared to when there is. Additionally we show that we only find a strong signature in the corresponding surface signal when the system is subject to surface friction. We then provide evidence, in Section 4, to show that this NAM-like signal is mainly caused by the modification of winds in the (extra-tropical) lower stratosphere and the inclusion of winds in the middle and upper stratosphere have almost no influence on the final state of the life cycle. In Section 5 we further discuss and interpret some of our findings before, in Section 6, we summarise the main conclusions of this paper.





## 2   Model and Basic states

All simulations are run with the simple dry dynamical core model BOB (Built on Beowolf, see Rivier et al. (2002) for details).
125  The model solves a spectral representation of the primitive equations in pressure coordinates with truncation at horizontal
wave number 85. The discrete vertical levels are distributed with constant spacing $\Delta z = 250$ m up to a height of $z = 60$ km,
where $z = -H \ln(p/p_0)$ is a log-pressure coordinate with scale height $H = 7.5$ km and reference pressure $p_0 = 1000$ hPa. To
minimise upper boundary effects we add 10 additional model levels between $z = 60$ km and $z = 82$ km, equally spaced in
pressure. Note that we are using a substantially higher vertical resolution than has typically been used in similar studies, since
130  we found in particular the details of the non-linear decay phase of the baroclinic life cycles to be sensitive to changes in $\Delta z$ for
values larger than about $\Delta z = 250$ m, as also further explained in Section 3.

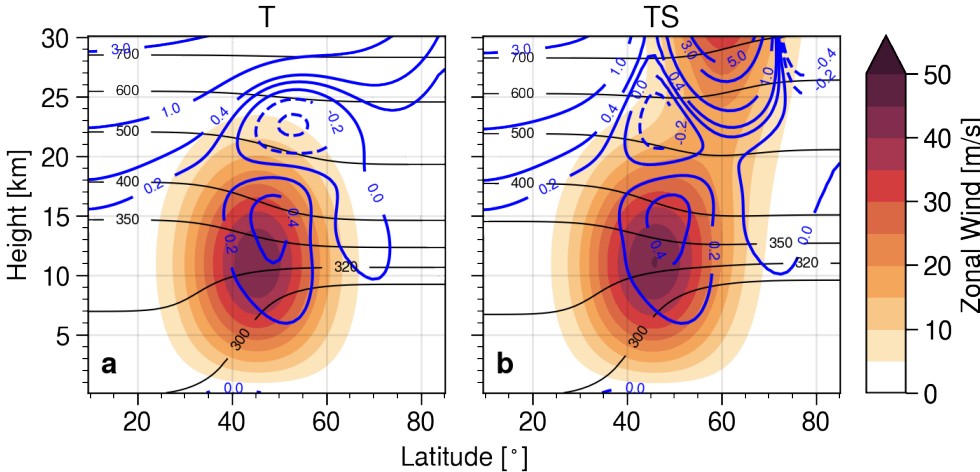

**Figure 1.** Examples of the basic state used in this study with different choice for parameter values to include either a) a tropospheric jet only
(experiment T) or b) a tropospheric and a stratospheric jet (experiment TS). The shading shows the zonal wind, thin black contours show
potential temperature [m/s] and thick blue contours show the meridional PV gradient [PVU/deg], with dashed contours corresponding to
negative values.

The model is initialised with a prescribed state and integrated forward in time with a step length of 5 minutes over a period
of 30 days, giving daily output of instantaneous fields (results are qualitatively unchanged for hourly output). We initialise all
experiments with an idealised and zonally symmetric basic state, loosely based on the initial state used by Kunz et al. (2009).
135  The basic state is analytically defined via a given zonal wind field and is chosen to represent two general situations, depending
on the choice of parameters: either a system with a tropospheric jet only (modelling post-SSW conditions), or a system that
contains a tropospheric and a stratospheric jet (mimicking more undisturbed winter-time conditions). In order to also study the
sensitivity of changes in the wind structure of different regions in atmosphere we further use a set of basic states which include
the tropospheric jet and only the upper or lower part of the stratospheric jet, respectively. Table 1 summarises the different





types of basic state configurations used in the present study. The two main basic state configurations (T and TS) are visualised in Figure 1 (note that only a part of the domain is shown).

**Table 1.** Different basic state configurations used.

| Experiment | Description |
|---|---|
| T | Tropospheric jet only |
| TS | Tropospheric and stratospheric jet (with magnitude $u_{Smax} = 75$m/s) |
| $TS_{<z_\eta}$ | Tropospheric and lower part of the stratospheric jet (below height $z_\eta$) |
| $TS_{>z_\eta}$ | Tropospheric and upper part of the stratospheric jet (above height $z_\eta$) |

The temperature distribution of to the respective initial state is calculated to be in thermal-wind balance with the prescribed wind field. Note that the resulting meridional PV gradient (thick blue contours in Figure 1) strongly depends on the vertical curvature of the underlying wind field and therefore produces a pronounced local maximum near the tropospheric jet core.

Further note that both configuration displayed in Figure 1, due to the strong dependency on the wind field structure, include regions with (slightly) negative PV gradient, which could potentially influence the evolution of the life cycle. However, the corresponding initial states follow the typical setup used in this type of idealised life cycle experiment. We further performed a series of sensitivity experiments and concluded the regions of negative PV gradient to have no significant influence on the qualitative results presented in this paper. Magnusdottir and Haynes (1996) also raised the question of the effect of negative PV gradients in typical life cycle setups on the evolution of the baroclinic wave and concluded that these regions can have an effect on certain details of the non-linear phase (e.g., details of the energetics), but seem to have no impact on most aspects of the qualitative behaviour.

To trigger the growth of a baroclinic wave the initial state is perturbed by super-imposing a zonally periodic near-surface temperature perturbation of fixed zonal wave number 6, centred around 45° latitude. We found our results to be qualitatively similar for perturbations with wave number 7, but the stratospheric jet to have almost no influence on the life cycle for wave numbers 5 and 8 (in these cases the purely tropospheric life cycle is generally weaker than for perturbations with wave numbers 6 and 7).

More details on how the basic state is constructed are given in the Appendix. Starting from the described initial conditions the experiments are then either run freely (without any external forcing) or including a linear Rayleigh surface friction, following the friction profile specified by Held and Suarez (1994) with a maximum friction coefficient of $k_f = 1$ day$^{-1}$ at the surface, gradually reducing to zero at 700 hPa ($z \approx 3$ km).

## 3 Modification of the life cycle by a stratospheric jet

We start our study by investigating in what way the general evolution of an idealised baroclinic life cycle is altered when the initial conditions include a tropospheric and a stratospheric jet, the latter representing the winter time polar vortex, compared





to when they include a tropospheric jet only, as is usually the case after a SSW and is the conventional life cycle setup. In the rest of this section we therefore analyse a set of life cycle experiments with varying values of the stratospheric jet strength parameter $u_{Smax}$ (see Equation A2 in the Appendix) and thus varying strength of the stratospheric jet that is added onto the system with tropospheric jet only.

### 3.1 Modification of the baroclinic wave breaking

The evolution of idealised baroclinic life cycles is often described in terms of the distribution of potential vorticity (PV) on an isentropic surface close to the jet core (or equivalently close to the tropopause). Zonal modulations in PV contours in this region of sharp PV gradient (also seen in Figure 1) give insights into the growth and decay of the eddy field, while any change in the position of the maximum in zonally averaged PV gradient represents a meridional shift of the jet. The top and middle rows in Figure 2 show the horizontal PV distribution on the 350 K isentrope at selected days for the two initial state configurations

with tropospheric jet only (experiment T) and tropospheric and stratospheric jet (experiment TS).

The general evolution of both experiments is similar to each other in the sense that the baroclinic wave grows gradually until about day 6. At that point the wave becomes non-linear, breaks and eventually decays. However, especially the non-linear decay phase shows substantial differences in the specific evolution of the PV field when a stratospheric jet is present. The wave breaking is still characterised by filaments of high PV that stretch out on the equatorward side of the jet core, break off and

eventually roll up anticyclonically, but the timing of events and the details of the small scale structures are altered considerably compared to the tropospheric jet only case. The decay of the baroclinic wave happens faster and at day 9 a new wave structure seems to have grown already, showing strong characteristics of cyclonic wave breaking (sometimes referred to as LC2 life cycle in contrast to the anticyclonic LC1 life cycle; see, for example, Thorncroft et al. (1993) for further details).

To highlight the modification in PV evolution induced by the presence of a stratospheric jet the bottom panel of Figure

2 shows the difference in the PV field of a simulation with and without stratospheric jet. Overlaid are the corresponding 8 PVU contours of the two respective experiments. It can be seen that at day 6, i.e., at the end of the linear growth phase, the two baroclinic waves have a similar magnitude and structure, but are slightly phase shifted with respect to each other. This shift can potentially be explained by a minor increase in phase speed in the case with a stratospheric jet. This might either be due to a minor increase in wind speed near the tropopause (also further discussed in Sections 4 and 5) or a change of

the corresponding PV gradient in that region. While a pure zonal phase shift of the wave should not have any influence on the subsequent behaviour of the wave-breaking due to the zonal symmetry of the system, it does indicate a change in the dispersion relation.

At days 7 to 9, i.e., during the non-linear phase, the evolution of the system is strongly influenced by the stratospheric jet and Figure 2 shows a large difference in PV distribution. Especially at days 8 and 9 the baroclinic wave in experiment TS,

including a stratospheric jet, seems to have entered a second growth phase, while the wave in experiment T still seems to be decaying. As mentioned in Section 1 these 'secondary life cycles' during the non-linear decay phase have been discussed previously by Barnes and Young (1992). We find the details of the non-linear phase, like the occurrence, timing or apparent flavour (in a LC1/LC2 sense) of 'secondary cycles', to be very sensitive to small changes of the initial conditions or the details




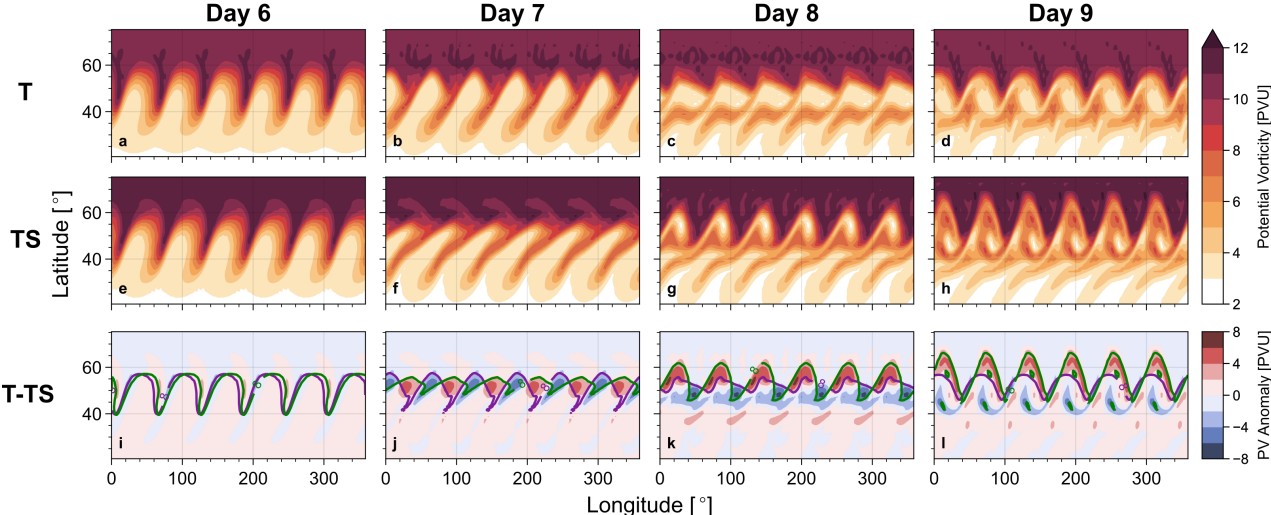

**Figure 2.** Evolution of PV distribution on the 350 K isentrope at different days for a system with tropospheric jet only (experiment T, top panel) or a tropospheric and stratospheric jet (experiment TS, middle panel). The bottom panel shows the difference of both experiments (T-TS), with 8 PVU contours of the respective full fields superimposed.

of the physical processes involved, as can also be seen in Figure 2. Recall that, as mentioned in Section 2, the occurrence and
strength of these secondary cycles varied in a set of sensitivity experiments with lower vertical resolution. For the purpose of
this study we therefore focus primarily on the evolution of the entire life cycle, e.g., in terms of the difference between initial
and some final state.

### 3.2   Dependency on stratospheric jet magnitude

In addition to the evolution of the PV field baroclinic life cycles can be quantified in terms of the global energetics of the system,
typically with a strong focus on eddy kinetic energy (EKE), which describes the growth and decay of the baroclinic wave in
the region of large meridional PV gradient near the jet core (see Figure 1). In particular the decay of EKE is associated with
an energy transfer to the zonal mean state, i.e., an increase of the mean kinetic energy (MKE). This increase in MKE can be
associated with a poleward shift, and a corresponding acceleration, of the tropospheric jet due to wave-mean-flow interactions
and poleward eddy momentum fluxes during the decay phase of the life cycle.

    The way the evolution of the life cycle is altered by a stratospheric jet can be seen in terms of EKE and MKE time series,
shown in Figure 3 for experiments with different values for the stratospheric jet magnitude $u_{Smax}$ (see Appendix for details).
Note that here we use $\Delta$MKE, which is simply the change in MKE with respect to the initial conditions and that both, EKE
and $\Delta$MKE, are displayed as vertically integrated and horizontally (over the northern hemisphere) averaged energy densities.

    In agreement with Figure 2, which suggests only a phase shift in the baroclinic waves during the linear phase, but no
difference in magnitudes, Figure 3 shows essentially no sensitivity to introducing a stratospheric jet before day 6, in particular





we do not find any significant change in growth rate, as has been reported by other authors, e.g., Wittman et al. (2007). A potential explanation for the strong change in growth rate found by Wittman et al. (2007) could be a substantial difference in meridional PV gradient (due to the substantial modification of the vertical curvature of zonal wind at the tropopause) between their different experimental setups. The basic states used in the present study, on the other hand, do only slightly differ in

terms of their tropopause level PV gradients (see Figure 1). However, during the non-linear-phase, so from day 7 onwards, the stratospheric jet seems to extensively alter the evolution of the life cycle. Especially the onset of a secondary phase of wave growth (with EKE peaking again at about day 10) seems to happen about a day earlier when a stratospheric jet is present in the system, and leads to a much stronger and more persistent secondary peak. The persistently elevated EKE of the secondary cycles during the non-linear phase (with EKE reducing again towards the final state) is consistent with the idea of a stronger

LC2 flavour (which is often characterised by persistently increased EKE in the decay phase) of the secondary cycles, as is also suggested by Figure 2 and is further discussed in Section 5.

The alteration of the system as we increase $u_{Smax}$ does not only manifest as changes in the details of how the wave breaking evolves, but also leads to a change of the final state (here defined as average over days 20-30), in particular a systematic increase of $\Delta$MKE.

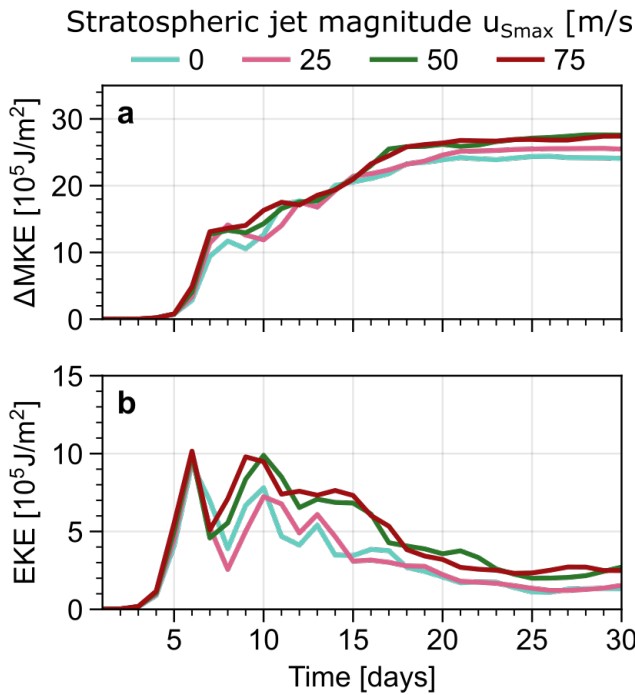

**Figure 3.** Evolution of mean kinetic energy change (a) and eddy kinetic energy (b) of the system with a tropospheric jet and a stratospheric jet with varying strength parameter $u_{Smax}$ (see Equation A2). The case with $u_{Smax} = 0$ corresponds to experiment T, the case $u_{Smax} = 75$ m/s to experiment TS. Energies are displayed as vertically integrated and horizontally averaged energy densities.

The elevated values of ΔMKE in the final state are linked to a stronger poleward shift (and correspondingly a stronger acceleration) of the tropospheric jet during the course of the life cycle when a stratospheric jet is present. This relative shift (compared to the experiment T, with tropospheric jet only) can be seen in Figure 4, which shows in all subplots as black contours the evolution of the zonal mean zonal wind field at 10 km. Figure 4a furthermore shows the zonal wind anomaly of experiment T with respect to the initial conditions. One can clearly see a dipole pattern developing around the initial jet core

(45° latitude) at the start of the non-linear phase at about day 6 and strengthening roughly until day 15, corresponding to a poleward shift of the jet core to about 60° latitude.

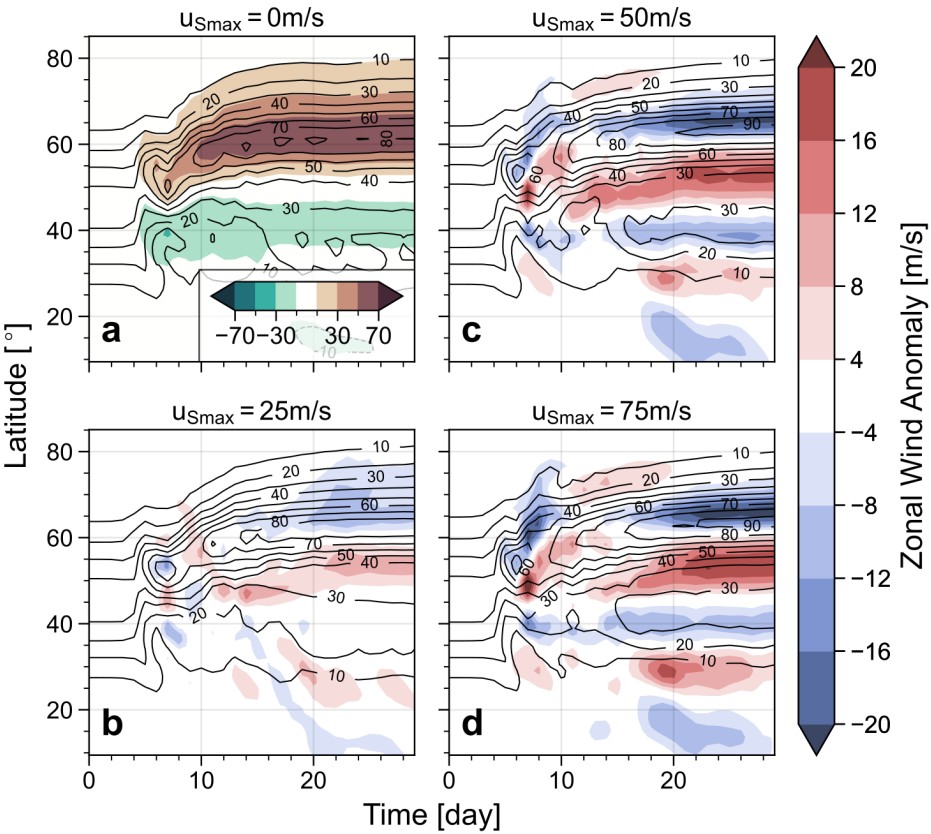

**Figure 4.** Black contours: Evolution of zonal mean zonal wind [m/s] on the 10 km surface for experiments with tropospheric jet only (a) or tropospheric and stratospheric jet of varying strength (b-d); the case with $u_{Smax} = 0$ corresponds to experiment T, the case $u_{Smax} = 75$ m/s to experiment TS. The shading in (a) shows the wind anomaly with respect to the initial state, the shading in (b-d) shows the wind anomaly induced when the stratospheric jet is removed from the system.

Figures 4b-d show the evolution of zonal mean zonal wind anomaly at 10 km of experiment TS, with varying strength of the imposed stratospheric jet, relative to experiment T, with tropospheric jet only. As suggested by the MKE time series shown earlier the zonal wind anomaly evolution indicates a dipole around the position of the final jet core emerging during the non-





linear phase of the life cycle. The change in zonal wind corresponds to a stronger poleward shift of the jet during the final

state in cases where a stratospheric jet is present, or equivalently, a relative equatorward shift of the tropospheric jet when the

stratospheric jet is removed. This jet shift is analogous to the NAM-like signature that has been observed after SSW events. It

further indicates the importance of tropospheric synoptic-scale eddy feedback in causing the observed negative NAM-signal,

as has previously been shown by other studies (e.g., Domeisen et al., 2013; Hitchcock and Simpson, 2014), and allows for a

simple way to quantify this eddy feedback.

### 3.3    Vertical structure of the response and influence of surface friction

The vertical structure of the relative jet shift of the final state can be seen in Figure 5, showing the difference in zonal mean

zonal wind during the final state (days 20-30 mean) between experiments T (with tropospheric jet only) and TS (also including

a stratospheric jet of magnitude $u_{Smax} = 75$ m/s). Subplot 5a shows the latitude-height equivalent of subplot 4d averaged

over the final state, while subplot 5b illustrates the corresponding zonal wind anomaly for an experiment with surface friction

applied to the system (see Section 2 for details). Both subplots show a clear equatorward jet-shift signature around the jet core

of the final jet when the stratospheric jet is removed.

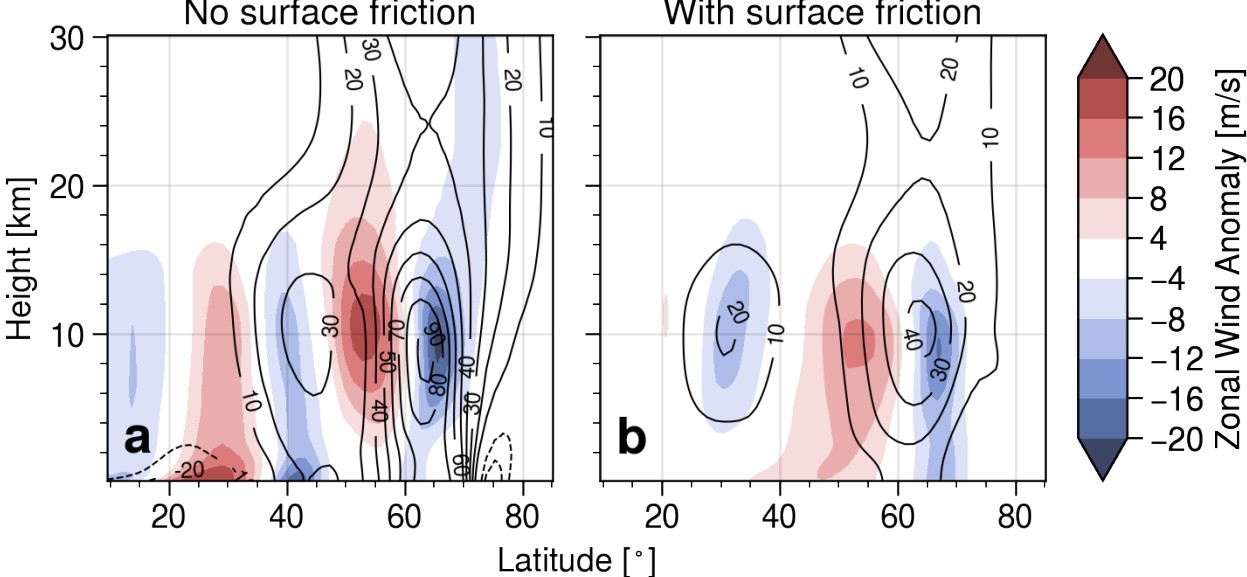

**Figure 5.** Contours: zonal mean zonal wind [m/s] of the final state (days 20-30 average) of a system with stratospheric jet. Shading: Changes
to the final state zonal mean zonal wind when the stratospheric jet is removed from the system. Subplot a) shows an experiment without
surface friction, while b) displays an experiment with surface friction as described in Section 2.

However, several differences can be observed. First, an overall weakening of the jet in the final state (black contours) can

be observed when surface friction is included, which is easily explained by the direct dissipation of kinetic energy over the

 

course of the life cycle due to the added friction process. The same argument holds for the disappearance of the strong wind anomaly patterns close to the surface at about 30° and 40° latitude in the case without friction. These patterns develop due to strong temperature fluxes in this region arising from the large meridional surface temperature gradient (see Figure 1) and they are likely not influential on the standard baroclinic life cycle evolution. More importantly, however, the vertical structure of the dipole pattern around the final jet core at 60° latitude is drastically different between the experiments with and without surface friction displayed in Figure 5. When the system is subject to surface friction during the life cycle the corresponding dipole pattern is more barotropic, thus it extends much further down and shows much stronger anomalies at the surface.

Figures 4 and 5 indicate a tendency of the tropospheric jet to exhibit a weaker poleward shift during the baroclinic life cycle if there is no stratospheric jet present compared to when there is. This behaviour is consistent with the negative NAM response, associated with an equatorward shift of the tropospheric jet, observed during periods following SSWs (see Baldwin and Dunkerton, 2001). It further provides a simple way to quantify the eddy feedback processes potentially involved in creating the corresponding jet shift signal. Figure 5 shows the shift signal only to have a significant surface contribution if the system is subject to surface friction.

To further illustrate the surface signal observed in our model experiments Figure 6 shows the geostrophic geopotential height field $Z$, calculated by solving the equation

$$\partial_\phi Z = -fa\bar{u} - \bar{u}^2 \tan\phi \tag{2}$$

via simple numerical integration with boundary condition $Z(\phi = 0) = 0$ for the zonal mean zonal wind field $\bar{u}$ of the final state. Here $f$ is the Coriolis parameter, $a$ the radius of the Earth, $g$ the gravitational acceleration and $\phi$ the latitude. Since $\bar{u}(z=0)$ vanishes for the initial state the surface geopotential height $Z_{sfc} \equiv Z(z=0)$ of the final state (or more precisely its gradient) describes the change in surface winds induced over the course of the baroclinic life cycle.

Figure 6 shows $Z_{sfc}$ for experiments that include surface friction and two different sets of initial conditions: T and TS, i.e., including a tropospheric jet only and including both, a tropospheric and a stratospheric jet. For both experiments we find the development of strong meridional gradients in $Z_{sfc}$ at around 50° or 60° latitude, respectively, consistent with strong surface winds. The farther equatorwards shifted position of the gradient of $Z_{sfc}$ in experiment T, relative to experiment TS, indicates again the relative equatorward shift of the final tropospheric jet if the stratospheric jet is removed from the initial conditions, corresponding to the NAM-signal discussed earlier.

The strength of the NAM-like jet shift signal depends on the magnitude of the stratospheric jet ($u_{Smax}$) included in the system, as can be seen in Figure 7. First, the NAM-signal, in form of a dipole jet shift pattern around 60° latitude, seems to develop for stratospheric jet magnitudes below about $u_{Smax} \lesssim 50$ m/s, but stays mostly unchanged for stratospheric jets exceeding $u_{Smax} \gtrsim 50$ m/s. Second, the NAM-response does not seem to be symmetric for positive and negative values of $u_{Smax}$. While the jet shift signal develops already for relatively weak westerly stratospheric jets, no coherent such signal can be observed for easterly stratospheric jets for the parameter range shown (a positive NAM-signal, i.e., a relative poleward shift, only starts to develop for $u_{Smax} \lesssim -40$ m/s).



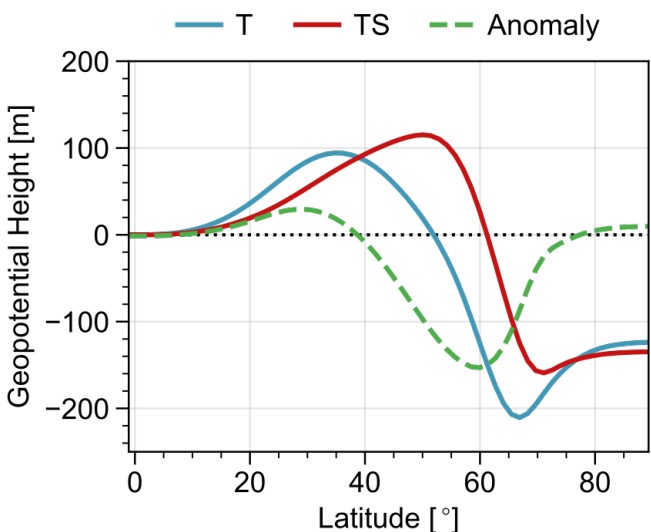

**Figure 6.** Zonal mean geopotential height at 1000hPa (or equivalently $z = 0$) of the final state for two experiments with surface friction and with tropospheric jet only (T) and tropospheric and stratospheric jet (TS), respectively. The dashed line shows the difference of both.

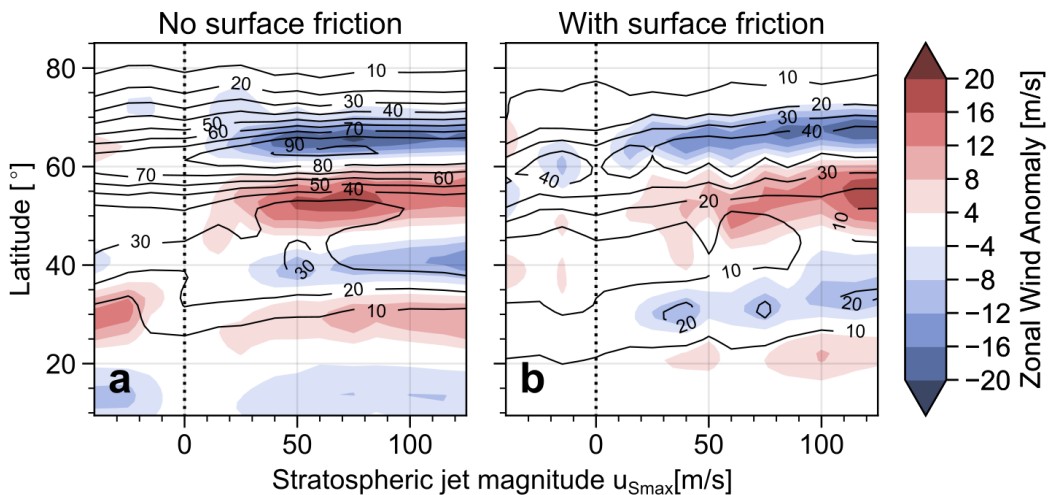

**Figure 7.** Contours: Zonal mean zonal wind at 10 km of the final state of experiments that include a stratospheric jet with varying strength parameter $u_{Smax}$. Shading: The changes induced when the stratospheric jet is removed from the system. The two subplots show experiment without and with surface friction, respectively. The vertical dotted line indicates $u_{Smax} = 0$, and thus experiment T.





In the rest of this paper we investigate the influence of a stratospheric jet onto the final state of the baroclinic life cycle and the resulting NAM-like signature in more detail. In particular we identify a region in the lower stratosphere which is highly
sensitive to changes in the zonal wind that are induced by the inclusion of a stratospheric jet.

## 4   Sensitivity of the life cycle to changes in the extratropical lower stratosphere

In the previous section we established that introducing a stratospheric jet can modify the evolution of the system in an idealised baroclinic life experiment, as has also been shown by other authors (e.g., Wittman et al., 2004). In this section we show that the system is particularly sensitive to changes in wind structure in the extratropical lower stratosphere (heights below about 25
km), while changes in the middle and upper stratosphere have almost no influence on the final state. In order to investigate this sensitivity we analyse a set of experiments with initial conditions that include a tropospheric jet, as well as a stratospheric jet with modified vertical structure.

We modify the structure by multiplying the profile of the stratospheric jet used in experiment TS by a function $\eta(z)$ (see Equation A2). We choose $\eta(z)$ to follow a tanh-profile, which allows us to smoothly set the winds of the stratospheric jet
component to zero below or above a set transition height $z_\eta$ and thus investigate which part of the stratospheric jet has the strongest influence on the life cycle. We hereby refer to the experiments where we only include the part of the stratospheric jet below height $z_\eta$ as '$TS_{<z_\eta}$', and correspondingly refer to the experiments where we keep the part above $z_\eta$ as '$TS_{>z_\eta}$' (for simplicity we drop the units of $z_\eta$ within this notation and set it to be kilometres). See the Appendix for details on how the basic state is defined.

Figure 8 illustrates the different basic states in terms of the full zonal mean zonal wind field, and the anomaly with respect to experiment T, i.e., the experiment without any superimposed stratospheric jet. Subplots 8a and b show experiments T and TS, including no or the full stratospheric jet, respectively. The experiments displayed in subplots 8b and c only superimpose the upper part of the stratospheric jet, above either 25 km or 10 km, while the experiments displayed in subplots 8e and f only include the respective lower parts.

Details of the vertical structure of the various initial wind fields can also be seen in Figure 9, displaying the zonal wind at $60°$ latitude, i.e., at the northern flank of the tropospheric jet and through the core of the stratospheric jet. A very prominent difference is that profiles where the stratospheric jet reaches into the lower stratosphere have substantially increased wind speeds in that region (roughly between 10 and 25km), compared to profiles where the contribution of the jet is mostly confined to the troposphere or the middle and upper stratosphere. This criterion divides the six profiles into two groups, 'Set 1' consisting
of profiles T, $TS_{>25}$ and $TS_{<10}$ with weak winds in the lower stratosphere, and 'Set 2' consisting of profiles TS, $TS_{>10}$ and $TS_{<25}$ with strong winds in the lower stratosphere. In most of the rest of this section we analyse the experiments with different initial conditions keeping in mind the grouping into these two sets.

To visualise the NAM-like jet shift signature of the final state, and to investigate which contribution to this jet shift can be associated the different parts of the stratospheric jet, Figure 10 shows the zonal mean zonal wind averaged over days 20-30 and
the corresponding anomaly from experiment T (with tropospheric jet only).



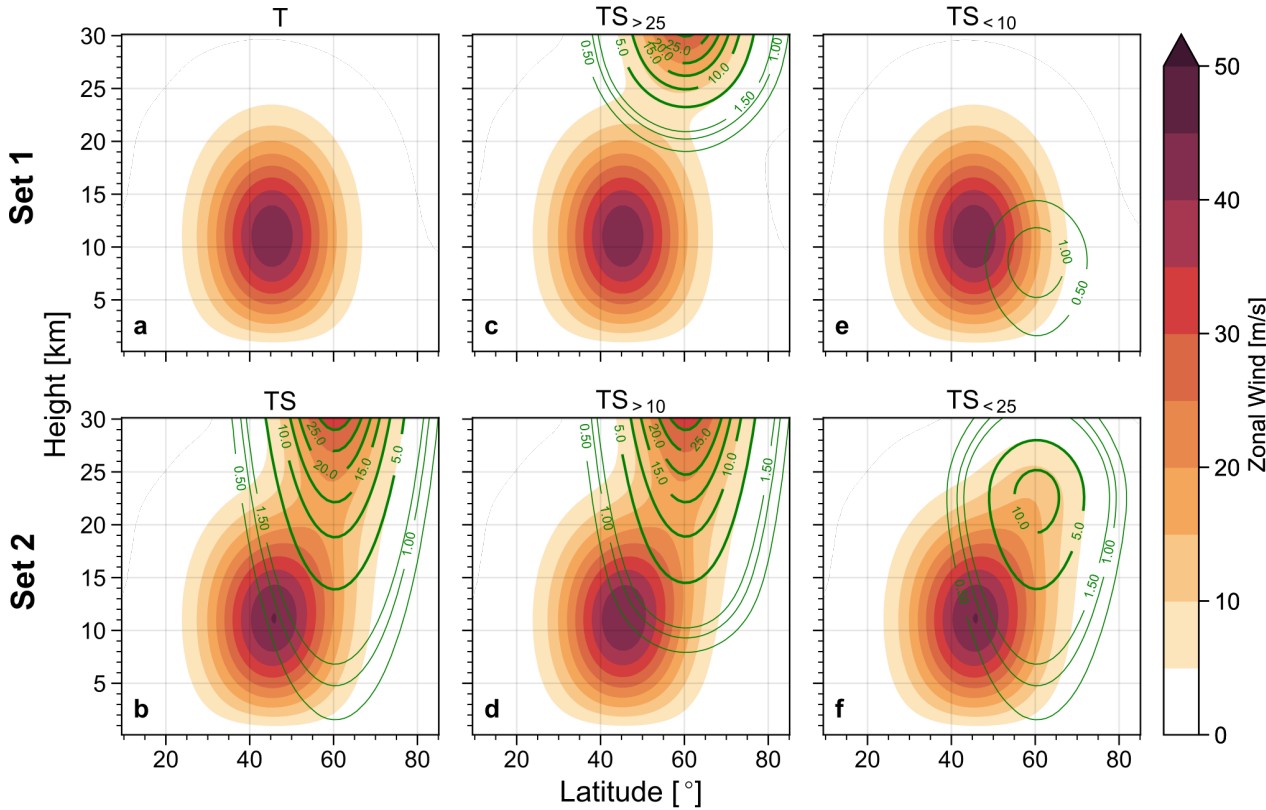

**Figure 8.** Zonal wind (shading) and anomaly from experiment T (green contours, in m/s) of the initial conditions for experiments with a tropospheric jet and varying vertical profiles of the superimposed stratospheric jet depending on the function $\eta(z)$ in Equation A2. Note that both sets of green contours, thick and thin, show the same quantity, but for different level ranges.

We first look at the experiments of Set 1. The final state zonal wind field of experiment $TS_{>25}$ (Figure 10c) does not show any substantial deviation from experiment T, indicating that winds in the middle and upper stratosphere have virtually no influence on the life cycle. Experiment $TS_{<10}$, with superimposed winds confined to the troposphere, shows a dipole pattern, which could potentially be attributed to the projection of the wind modification on e.g., the increase in tropospheric jet magnitude or

the vertical shear, also further discusses in Section 5. However, also note that the superimposed winds of the stratospheric jet do not abruptly vanish at the given cut-off height (e.g., above 10 km for $TS_{<10}$), but follow a smooth transition over the course of about 4 km and therefore still reach into the lower stratosphere region.

The experiments of Set 2 (bottom panel of Figure 10) do all show a clear dipole structure in the anomaly field, centred around about 60° latitude. Note in particular the strong signal of experiment $TS_{<25}$, where the superimposed winds are confined to the

troposphere and lower stratosphere, further suggesting the winds in the middle and upper troposphere to have no significant contribution in causing the observed jet shift. Experiment $TS_{>10}$, including a stratospheric jet that reaches into the lower



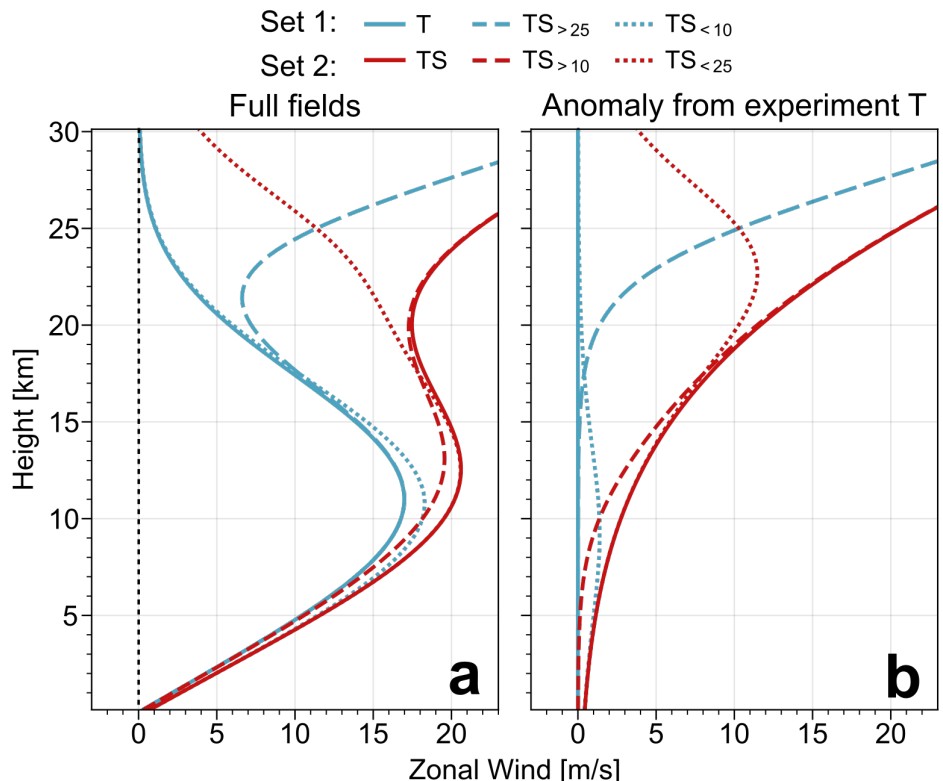

**Figure 9.** Vertical profiles of zonal mean zonal wind at $60°$ latitude of the initial conditions for different experiments without (T), with full (TS) or with partial stratospheric jet (other profiles). Subplot a) shows the full fields, subplot b) the anomaly from experiment T.

stratosphere but does reach not far into the troposphere, also shows a clear dipole pattern in zonal wind anomaly. In particular compare experiments $TS_{>10}$ and $TS_{>25}$, as well as $TS_{<25}$ and $TS_{<10}$: in both cases does the jet shift signal increase in strength when the superimposed stratospheric jet reaches into the lower stratosphere (10 km to 25 km), compared to when it does not.

The significance of the lower stratospheric wind anomalies are discussed further in Section 5.

The surface signal of the NAM-like response discussed above can be seen in Figure 11, displaying the zonally averaged geostrophic geopotential height field calculated via Equation 2. It can clearly be seen how the different experiments show indications for NAM-like surface signals in good agreement with what is shown in Figure 10. Especially the experiments of Set 2 (bottom panel) show a poleward shift and acceleration of the surface winds (in terms of gradient of the shown curves)

relative to the reference experiment T, with only tropospheric jet.

Figure 11b further shows the sum of the geopotential height anomalies induced by removal of the (partial) stratospheric jet from the experiments $T_{<10}$ and $T_{>10}$, i.e, experiments where we only include the part of the stratospheric jet above or below



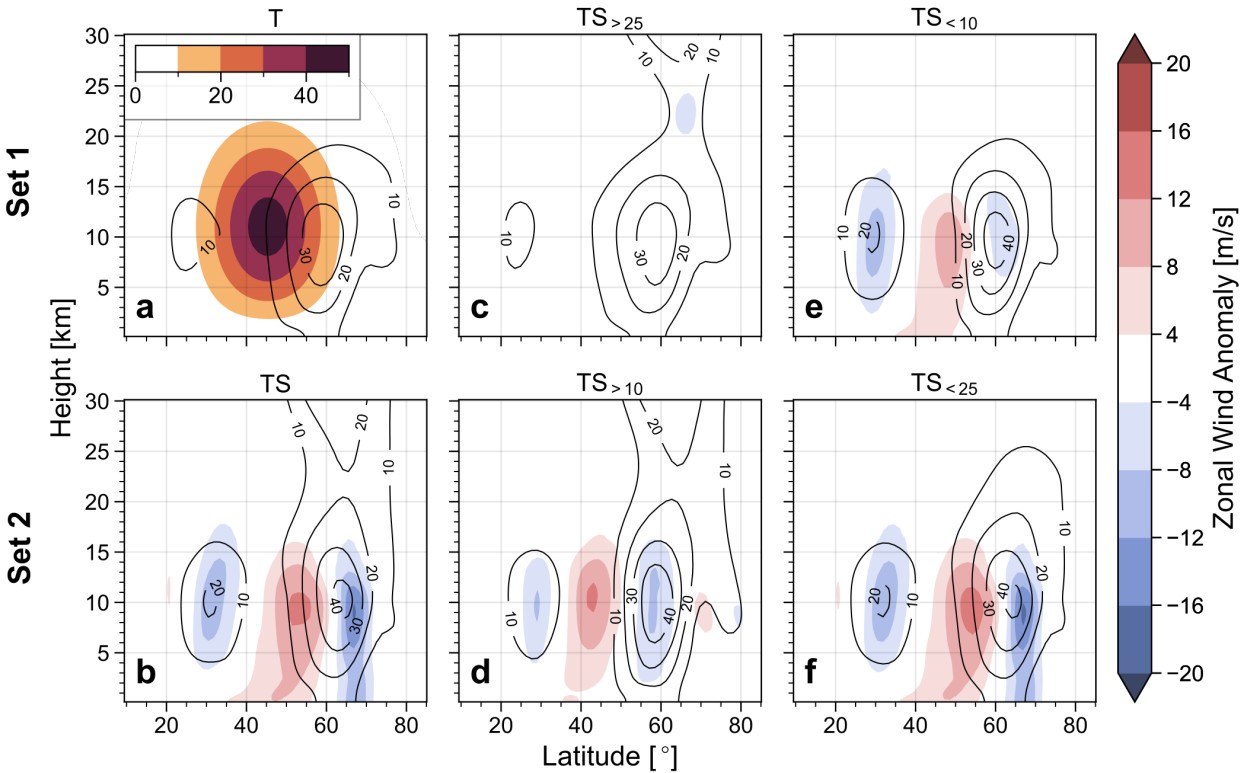

**Figure 10.** Black contours show the zonal mean zonal wind [m/s] distribution of the final state for different experiments. The shading in (a) indicates the initial zonal winds [m/s] of experiment T, the shading in (b-f) shows the anomaly from experiment T in zonal mean zonal wind of the final state. All experiments include surface friction.

10km. The similarity of this sum to the corresponding geopotential height anomaly of experiment TS, with full stratospheric jet included, suggests a certain additivity of the response to the stratospheric jet[1], also further discussed in Section 5.

## 5   Discussion

In Section 3 we found the presence of a stratospheric jet to substantially alter the non-linear decay phase of a baroclinic life cycle. In particular Figures 2 and 3 showed changes in the secondary cycles occurring during the decay stage, including changes in number, strength, duration, timing and apparent type (or flavour) of these secondary cycles. The different types of baroclinic wave breaking (LC1 and LC2) have been linked to different weather regimes, and thus corresponding transitions within a life 350   cycle can potentially have a large impact on surface weather (e.g., Michel and Rivière, 2011). As discussed, the baroclinic decay phase of experiment TS shows characteristics of both, LC1 and LC2 flavour, or equivalently cyclonic and anticyclonic

---

[1]Note that the same similarity seems to also hold for the sum of anomalies of experiments $T_{<25}$ and $T_{>25}$ (not shown explicitly).

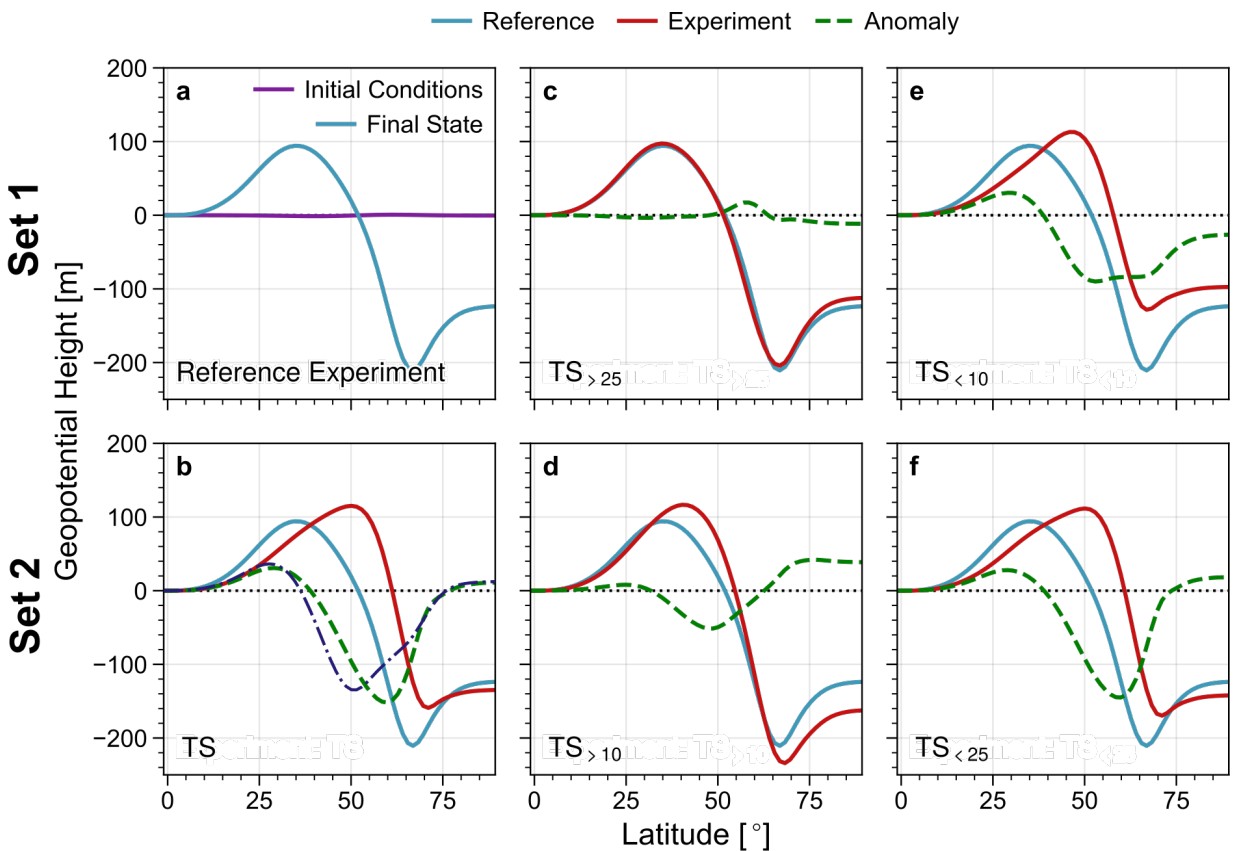

**Figure 11.** Geopotential height at 1000 hPa of the final state of the experiments displayed in Figure 8. Subplot a) illustrates the difference in initial and final state geopotential height for experiment T, all other subplots illustrate the difference between the respective experiment and the reference experiment T. The purple dash-dotted line in subplot b) shows the sum of the anomalies for the cases $T_{<10}$ and $T_{>10}$. All experiments include surface friction.

wave breaking, while experiment T shows more LC1 characteristics. However, the general behaviour of the TS life cycle (e.g., in terms of its final state) still follows primarily the (anticyclonic) LC1 type, and it only seems to experience individual, transient (cyclonic) LC2 wave-breaking events. The importance of these transient LC2 events for the overall meteorological
regime is presently not clear. Note that other authors have previously also reported about transitions between LC1 and LC2 wave breaking states based on stratospheric conditions (e.g., Kunz et al., 2009), but mostly in terms of the entire life cycle, rather than in terms of more transient events.

The introduction of a stratospheric jet not only modifies the details of the wave breaking during the decay phase, but also the (steady) final state of the life cycle. In particular we observed a relative equatorward shift of the tropospheric jet in the
final state when removing the stratospheric jet from the initial conditions of the system, as can be seen in Figures 5 and 6. This





jet shift is analogous to the NAM-like signature that has been observed after SSW events. To what extent the observed NAM response to SSWs is similarly influenced by tropospheric eddy feedbacks, as suggested by our results, remains to be quantified further.

It is worth to point out that the relative meridional shift between the final jet in the experiments T and TS results from differences in the meridional eddy momentum transport during the life cycle (not shown). The increased momentum fluxes in experiment TS, compared to T, can be related to increased wave activity around tropopause level, which is consistent with the increase in EKE shown in Figure 3.

Figure 5 further shows that the surface signal of the NAM-response in the final state, given by the zonal mean zonal wind difference between experiments T and TS, is enhanced when the system is subject to surface friction. The effect of surface friction to increase the surface wind signal of the NAM-response might seem counter-intuitive. However, as we already pointed out in Subsection 1.3, surface friction can provide a way for tropopause level eddy momentum fluxes to couple to the surface winds. The modification of the baroclinic eddy field by the presence of a stratospheric jet can therefore project more strongly onto the surface winds and produce a stronger surface signal. The evolution equation of the vertically averaged zonal mean zonal wind (Equation 1) is often used to argue that on long time scales (where $\partial_t[\bar{u}] \approx 0$) the eddy flux convergence has to be balanced by the (dissipation of) surface winds. In our (transient) life cycle experiments we cannot neglect the wind tendency term and the main balance is given by $\partial_t[\bar{u}] \approx -\partial_y \left[\overline{u'v'}\right]$. However, the dissipation term $\bar{u}_{sfc}/\tau$ provides an important contribution to the equation and strongly modifies the acceleration of the jet, as is already suggested by the factor 2 difference of the final jet magnitude in the cases with and without surface friction (see Figure 5).

When interpreting Equation 1 one also has to keep in mind that it describes the evolution of the full (vertically and zonally averaged) wind field, whereas we are mostly interested in the enhanced surface signal of the difference in wind field between experiments TS and T (see Figure 5), i.e., the NAM-like shift signal. The line of argumentation, however, is analogous. The introduction (or removal) of a stratospheric jet influences the evolution of baroclinic eddies at tropopause level. Following Equation 1, the corresponding changes in eddy momentum flux then induce changes in the wind tendency (which will tend to be close to the level of the eddy flux, primarily near the tropopause), but also couple directly to the surface winds.

The enhancement of the surface signal by surface friction can potentially be understood via the following mechanism: the decay stage of the life cycle is characterised by a barotropisation of the tropospheric jet and thus a reduction of vertical shear and a strengthening of surface winds. The surface friction, on the other hand, tends to increase the vertical wind shear and acts as a source of baroclinicity. This increase in baroclinicity then leads to additional (secondary) life cycles during the late stages of the life cycle (seen around days 16-25 in Figure S2 in the supplementary material) and thus an additional barotropisation of the jet, enhancing the downward propagation of the jet-shift signal.

Figure 7 indicated that the strength of the NAM-response following the removal of the stratospheric jet depends non-linearly on the magnitude of the stratospheric jet. In particular, the signal seems to saturate when the stratospheric jet magnitude exceeds a certain value and stronger jets do not lead to a stronger NAM-signal any more. This behaviour suggests that an anomalously strong polar vortex does not necessarily lead to anomalously positive NAM-signals. Similarly Figure 7 indicates that a reversal of the stratospheric jet (with $u_{Smax} < 0$) does not lead to a negative NAM-response with respect to experiment T, which





suggests that in terms of NAM-response it is not important weather a SSW leads to slightly or strongly reversed winds of the polar vortex. However, the setup of baroclinic life cycle experiments does, of course, not capture the dynamics around SSWs in their entire complexity and these results do not necessarily carry over to the real atmosphere.

In Section 4 we showed that the NAM-response observed in the final state of our life cycle experiments is mostly caused
by the change in wind structure in the lower stratosphere when including the stratospheric jet, rather than wind anomalies in the middle and upper stratosphere (see Figure 10). However, it should be noted that changing the wind structure in the lower stratosphere does also introduce changes in various other characteristics of the corresponding initial conditions, like the height of maximum wind speed, the vertical wind shear in the upper troposphere (roughly up to 10km) and the magnitude of the tropospheric jet (especially obvious for profiles T and TS in Figure 9). However, these three characteristics are intrinsically not
completely independent and can potentially all affect the evolution of the life cycle. This can be seen, e.g., since the vertical wind shear is (via thermal wind balance) related to the horizontal temperature gradient, which drives the growth of baroclinic waves and can, among other things, modify their (linear) growth rate (although note that the near surface shear is almost identical in the different experiments).

We performed a set of sensitivity experiments (not shown) with tropospheric jet only and varying tropospheric jet magnitude
(and therefore increased vertical shear in the troposphere). We found that an increase in tropospheric jet strength also leads to a increased poleward shift during the life cycle (i.e., an equatorward shift of the jet in the final state of experiment T relative to a case with stronger tropospheric jet), similar to the shift observed in Figure 5a. In order to achieve a jet shift signal of similar magnitude as the one shown in Figure 5, however, it was necessary to increase the jet magnitude by order of 10 m/s (the difference in tropospheric jet magnitude between experiments T and TS is only of the order of 1 m/s.), indicating that
other characteristics of the initial state need to contribute and the observed jet shift cannot purely be a result of a strengthened tropospheric jet. The inclusion of the stratospheric jet does to some extend project onto the mentioned characteristics (e.g., height of the jet core and tropospheric shear) of the total zonal wind profile and the resulting jet shift can potentially be interpreted as the result of a combination of factors.

Figure 11 further suggested that we essentially recover the surface geopotential height signal of experiment TS (with full
stratospheric jet), when adding the corresponding signals of experiments $T_{<10}$ and $T_{>10}$. Such additivity of responses might be another indication that the stratospheric jet projects onto various other structures and characteristics (e.g., tropospheric shear and jet core height) and the corresponding jet shift response forms as the result of a combination of responses to those modifications. However, while the anomalies of the respective experiments seem to be additive when it comes to the surface geopotential height, the middle tropospheric jet shift response in Figure 10 does not appear to follow the same additive behaviour.

As discussed, Figure 10 suggests the NAM-like jet shift signature of the life cycle due to the inclusion of a stratospheric jet to be mainly caused by the corresponding change in winds in the lower stratosphere, rather than the winds in the middle and upper stratosphere, where the stratospheric jet itself is strongest. A similar conclusion can be drawn from the energetics of the system (provided as supplementary material), which shows a consistent increase in MKE of the final state for the experiments in Set 2 (as defined in Section 4), compared to the experiments in Set 1, in a system that does not include surface friction. As
also explained in Section 3, this increase in MKE is caused by the relative meridional shift of the final tropospheric jet. Note





that if the system includes surface friction the constant dissipation of winds leads to a gradual and flow dependent decrease of MKE, which makes the interpretation of the energetics in terms of a final state difficult.

## 6 Summary and conclusions

In this paper we discussed changes in the evolution of idealised baroclinic life cycles induced by the presence of a stratospheric jet. Particular focus was given on a jet shift signal in the zonal wind anomaly of the final state of the life cycle, similar to the signature of negative (surface) anomalies of the northern annular mode (NAM) often observed after sudden stratospheric warming (SSW) events.

We found that the final state of the life cycle is associated with increased zonal mean kinetic energy when a stratospheric jet is included in the system, roughly representing the polar vortex of typical winter-time conditions, compared to the typical life cycle setup including only a tropospheric jet, roughly representing post-SSW conditions. This increase in mean kinetic energy corresponds to a negative NAM signal in the final state zonal wind, i.e., a relative equatorward shift of the tropospheric jet in the case with tropospheric jet only compared to the case with tropospheric and stratospheric jet. The negative NAM signal is the result of a reduced poleward shift over the course of the life cycle induced by a reduction in eddy momentum transport at tropopause level.

The corresponding NAM-like jet-shift response has an increased surface signal if the system includes surface fiction, which might seem counter-intuitive, but is consistent with the idea of an increased coupling of surface winds to the eddy momentum transport at tropopause level due to the friction.

We further showed that the system is mainly sensitive to changes of the wind structure in the lower stratosphere (heights between 10 km and 25 km), rather than to zonal wind anomalies in the middle and upper stratosphere.

The findings of this paper improve our basic understanding of the weather and climate system in the mid-latitude troposphere and lower stratosphere. In particular they provide a potential explanation for the downward propagation of zonal wind anomalies from the stratosphere into the troposphere and the related negative NAM signal observed after SSWs. The idealised lice cycle setting further provides a quantitative way to analyse the importance of the tropospheric eddy feedback during and after SSWs.

## Appendix A: Appendix: Construction of initial state

The basic state used to initialise our experiments is defined via a zonally symmetric zonal wind field, consisting of two individual components: a tropospheric jet $U_T$ (representing the mid-latitude jet) and a stratospheric jet $U_S$ (representing the polar vortex). The total wind field is then given by the sum of both components $U = U_T + U_S$, with the tropospheric jet profile being given by

$$U_T = u_{Tmax} \left( z/z_{Tmid} \right) \exp\left( \left( 1 - \left( z/z_{Tmid} \right)^\alpha \right)/\alpha \right) \sin^3 \left( \pi \sin^2 \left( \phi \right) \right), \tag{A1}$$





where $z = -H \ln(p/p_0)$ is a log-pressure coordinate with scale height $H = 7.5$ km and reference pressure $p_0 = 1000$ hPa and $\phi$ describes latitude. The parameters $u_{Tmax}$, $z_{Tmid}$ and $\alpha$ can be used to modify the jet strength, the core height and the depth of the jet, respectively. The corresponding stratospheric jet profile is defined via

$$U_S = u_{Smax}\eta(z)\exp\left(-(z - z_{Smid})^2/\Delta z_S^2 - (\phi - \phi_S)^2/\Delta \phi_S^2\right), \tag{A2}$$

where $u_{Smax}$ determines the strength of the jet, $z_{Smid}$ and $\phi_S$ its core position and $\Delta z_S$ and $\Delta \phi_S$ its width and depth, respectively. Note that we restrict both jet profiles to the northern hemisphere, i.e., for $\phi < 0$ we choose $u_{Tmax} = u_{Smax} = 0$ and therefore keep the southern hemisphere of the basic state at rest.

The function $\eta(z)$ can be used to further modify the vertical structure of the stratospheric jet. For all experiments in Section 3 we choose $\eta \equiv 1$, so the stratospheric jet is unmodified, while for the cut-off experiment in Section 4 we choose

$$\eta(z) = 0.5\left(1 \pm \tanh\left((z - z_\eta)/\Delta z_\eta\right)\right), \tag{A3}$$

in order to set the stratospheric jet strength to zero above or below (depending on whether a plus or minus is used within Equation A3) the transition height $z_\eta$ with a smooth transition of depth $\Delta z_\eta$. This gives us a way to isolate the parts of the stratospheric jet within the troposphere, lower stratosphere or middle and upper stratosphere, respectively, and thus study the corresponding influence on the life cycles individually.

From this initial wind field we compute the meridionally varying part of the initial temperature field following the thermal wind balance approach used by Polvani and Esler (2007). The meridionally constant part of the (potential) temperature field is specified by the profile $\theta(z)$, which is constructed by solving Equation A4 for given (horizontally constant) static stability $N^2$ and surface potential temperature $\theta_{sfc}$.

$$N^2(z) = (g/\theta)\partial_z \theta, \tag{A4}$$

with gravitational acceleration $g$. The imposed profile of $N^2(z)$ is defined by Equation A5 and consists of two regions of constant static stability ($N_T^2$ and $N_S^2$, corresponding to troposphere and stratosphere) with a smooth transition at height $t_{trop}$.

$$N^2(z) = N_T^2 + 0.5(N_S^2 - N_T^2)\left(1 + \tanh\left((z - z_{trop})/\Delta z_{trop}\right)\right) \tag{A5}$$

To trigger wave growth due to the baroclinic instability of the system we perturb the temperature field of the initial state with a vertically and meridionally confined and zonally periodic disturbance of fixed zonal wave number $k$. The spatial structure
$T_{pert}$ of this temperature perturbation is defined via Equation A6. Following Polvani et al. (2004) we do not introduce an





equivalent balanced wind perturbation as the small imbalance of this initial perturbation only has a negligible effect on the general evolution of the flow, compared to the rapidly growing unstable modes of the system.

$$T_{pert} = T_{max} \cos{(k\lambda)} \cosh{(2 (\phi - \phi_{pert}))}^{-2} \exp{((p - p_0)/(p_0 - p_{pert}))}, \tag{A6}$$

where $p_0 = 1000$ hPa and $\lambda$ is longitude. Table A1 lists the physical parameters and parameter ranges used to define the
different basic states used in the present paper.

*Author contributions.* PR produced the idealised model simulations, analysed the corresponding output, produced the visualisations and wrote the paper. TB advised PR throughout this work, contributed to the interpretation of the results and improved the paper for the final version.

*Competing interests.* The authors declare that they have no conflict of interest.

*Acknowledgements.* This study was funded by the Transregional Collaborative Research Centre SFB/TRR 165 'Waves to Weather' of the German Research Foundation (DFG).



**Table A1.** Physical parameters used in the the different model experiments.

| Symbol | Physical meaning | Value |
|---|---|---|
| $u_{Tmax}$ | Tropospheric jet strength | 45 m/s |
| $z_{Tmid}$ | Tropospheric jet core height | 11 km |
| $\alpha$ | Tropospheric jet depth parameter | 3 |
| | | |
| $u_{Smax}$ | Stratospheric jet strength | 0-75 m/s |
| $z_{Smid}$ | Stratospheric jet core height | 50 km |
| $\Delta z_S$ | Stratospheric jet depth | 22 km |
| $\phi_S$ | Stratospheric jet core latitude | 60° |
| $\Delta\phi_S$ | Stratospheric jet width | 12° |
| | | |
| $\theta_{sfc}$ | Surface potential temperature | 288 K |
| $N_T^2$ | Tropospheric static stability | 1.2e-4 s$^{-1}$ |
| $N_S^2$ | Stratospheric static stability | 5e-4 s$^{-1}$ |
| $z_{trop}$ | Reference tropopause height | 12.5 km |
| $\Delta z_{trop}$ | Reference tropopause depth | 3 km |
| | | |
| $T_{max}$ | Temperature perturbation magnitude | 1 K |
| $k$ | Zonal perturbation wave number | 6 |
| $\phi_{pert}$ | Perturbation latitude centre | 45° |
| $p_{pert}$ | Perturbation pressure top | 700 hPa |
| | | |
| $z_\eta$ | Cut-off transition height | 10 km and 25 km |
| $\Delta z_\eta$ | Cut-off transition depth | 4 km |





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
