# Peer review of "Tropospheric eddy feedback to different stratospheric conditions in idealised baroclinic life cycles"

_Weather and Climate Dynamics, 2020_

## Referee Comment (RC1) · Anonymous Referee #1 · 7 Sep 2020

Tropospheric eddy feedback to different stratospheric conditions in idealised baroclinic life cycles

Philip Rupp and Thomas Birner

The authors report on a set of baroclinic lifecycle experiments aimed at better understanding the influence of the stratospheric vortex on tropospheric jets. Consistent with previous works, they find that life cycles are substantially modified by presence or lack of a stratospheric vortex. They point out the importance of the winds in the lower stratosphere and identify an amplifying effect of surface friction on the near surface response.

[Figure]

The experiments are interesting and potentially quite informative - more idealized experiments such as these are, in my opinion, under-utilized for studying this difficult problem. In particular I found them to be a very clear illustration of the importance of non-linear phase of baroclinic lifecycles in understanding the nature of the tropospheric eddy feedbacks. However, I found some of the discussion to be a bit unsatisfying. I think the most useful suggestion I can give is to strengthen and clarify the comparison of the present results with previous studies on baroclinic lifecycles. Section 1.2 does set up some past results, but it's never made that clear how the present results fit in. For instance: the Wittman papers emphasize changes in linear growth rates, while the present paper emphasizes the non-linear decay phase as the key period of difference, finding (from what I can tell) relatively weak changes in the growth rates. Is this discrepancy consistent with the Smy and Scott emphasis on PV in the sub-vortex region? How much does this have to do with the different basic states?

To be clear - overall this is a solid peice of science and should ultimately be worthy of publication. I have some further specific comments and suggestions below; some of which might involve significant further efforts. I do not require all of these suggestions to be pursued, but I do think the paper would be substantially improved if this main point can be substantially addressed.

Major comments/suggestions

PV structure

The PV structure of the initial conditions is emphasized in Fig. 1, but it is not really made reference to later in the interpretation. For instance, how do the perturbations considered in Section 4 modify the PV gradients? One clear difference between T and TS is the presence of narrow region of positive pv gradient near 20 km at 60 degrees in the presence of a stratospheric jet; is it really the winds in the lower stratosphere that are key, or is it the PV gradient structure? Moreover, how is the zonal PV structure changed over the evolution of the lifecycles?

[Figure]

**Energetics**

It seems that one way of interpreting these results, consistent with the discussion of Barnes and Young (1992) in Section 1.2, is that the presence of the stratospheric jet allows for a greater conversion of APE to EKE and ultimately MKE; and that this is further enhanced by including surface friction. If this is a fair characterization of the present results, how does the available potential energy differ in the various cases? Could this result be simply explained by the presence of more APE in the cases with a stratospheric vortex?

A related comment: in the presence of surface friction there isn't really final steady state as MKE will continue to dissipate. This is very briefly discussed right at the end of the discussion (l430), but until the reader gets there it's not really clear how the authors defined the final state. Would it be possible to put the figures in the supplementary information on more equal footing by computing the energy lost to surface frection in the runs in Fig. S2, showing instead the net EKE and MKE generated by conservative processes?

**Mass fluxes**

There is some discussion of the 1000 hPa 'Geopotential height' anomalies as a measure of the surface NAM or AO signature of the baroclinic adjustment. One substantial point is that equation (2) on line 270 assumes gradient wind balance holds near surface. This is ok for case with no friction, but not for the case with surface friction, which will modify dominant the force balance. This could be corrected by adding in surface friction on right hand side.

But I wonder if it might be more directly useful to consider the evolution of the surface pressure, and the nature of the mass fluxes in these experiments; in particular I wonder if these are substantially modified by the presence of an Ekman layer in the cases with surface friction.

Specific/Minor Comments:

l76: perturbation to what?

l188: could this alternatively be explained by greater meridional shear in TS simulation?

l213: Are EKE and MKE integrated over depth of troposphere or the depth of atmosphere? I would think that the MKE of polar jet is small but not negligible.

l310-317: I found the grouping of these experiments to be initially quite confusing, thinking that the natural ordering would be to group TS>10 with TS<10, and TS>25 with TS<25. I did ultimately figure out that the logic is intended to emphasize the wind anomalies in the lower stratosphere (to be fair, this is stated in the text, but preconceived notions can be hard to shake sometimes).

This is ultimately just an issue of presentation. But I did wonder if it would be clearer to readers to start with (what seems to me) the more natural a priori sorting instead of anticipating the results? If not, perhaps naming the sets something more descriptive, such as 'Weak winds/Strong winds' instead of 'Set 1/Set 2'?

l325 discussed

Fig. 7: How does u at 15 km vary over this set of integrations? This might be more illuminating than the mid-stratospheric winds.

l413-414: what is this statement based on? explicit calculations or assumption of linearity?

l423: The additivity of surface winds also is not perfect.

l450: This final paragraph also contributed to my feeling unsatisfied by the discussion. What is the explanation for the downward influence suggested by the present results? How can we use these experiments to quantify the eddy feedback? These aren't unreasonable claims, but the authors should explain them more explicitly.

---

## Referee Comment (RC2) · Anonymous Referee #2 · 11 Oct 2020

In this manuscript, the authors use an idealized model to investigate the role of the stratospheric polar vortex on influencing the tropospheric jet. Specifically, they test the sensitivity of the NAM-like equatorward jet shift associated with a weak or reversed stratospheric polar vortex to the height of the vertical winds in the polar jet, and to the magnitude of surface friction. They find that the tropospheric response is sensitive to changes in winds in the lower stratospheric polar vortex, but not to winds in the upper stratosphere. Additionally, they find that surface friction enhances the tropospheric response to stratospheric vortex changes, and that friction acts to bring nearly barotropic anomalies all the way down to the surface.

[Figure]

Although the findings in general agree with previous studies, the consolidation of the impacts of surface friction and lower-stratospheric anomalies using a coherent model framework produces a compelling standalone study with implications for our dynamical understanding of sudden stratospheric warmings. The manuscript is well-written, well-organized, and the scientific questions well-constructed, and I only have a couple minor comments and questions. Most of my corrections and minor questions can be found in the attached manuscript.

The main additional question I have is if the authors think this study can provide insight into the observed differences in timing between "Displacement" type sudden stratospheric warming events, and "Split" type sudden stratospheric warming events. Specifically, Splits are observed to show an almost-instantaneous NAM-like response to SSWs, whereas for Displacements, the response tends to occur with a significant lag on the order of weeks. Could this be related to a difference in the vertical structure of the polar vortex in Displacement versus Split events? For example, is it possible that Displacement events show anomalies that start higher up in the stratosphere and work their way downwards, while Split events produce a strong signal in the lower stratosphere almost immediately? I am not sure why or how surface friction could play a role, since I can't imagine any way in which surface friction could depend on the type of vortex breakdown.

Please also note the supplement to this comment:
https://wcd.copernicus.org/preprints/wcd-2020-35/wcd-2020-35-RC2-supplement.pdf

———————————————————

---

## Referee Comment (RC3) · Anonymous Referee #3 · 15 Oct 2020

**Overview**

      This paper posed an interesting scientific question regarding stratosphere-troposphere coupling, and addressed it with well-designed experiments. I think the authors' use of a more simplified atmospheric model was a good choice here, as these simplified simulations are able to capture most of the atmospheric dynamics relevant to their scientific interests. The experiments described in this paper explore the tropospheric response to the presence or absence of a stratospheric jet, in an idealized attempt to compare conditions that typically occur in wintertime (a strong stratospheric jet), and conditions that are common after a sudden stratospheric warming event (a much weaker or absent stratospheric jet). The authors explore a robust parameter space in their experiments, and they analyze a variety of stratospheric jet strengths, several different vertical structures of stratospheric winds, and the effects of surface friction. The authors find that when a stratospheric jet is absent, the tropospheric jet is located equatorward of where it is located in the presence of a stratospheric jet. The presence of surface friction enhances this response, with an even stronger equatorward shift of the tropospheric jet when surface friction is present and a stratospheric jet is absent. Finally, the authors note that the winds in the lower stratosphere are key for influencing the tropospheric response; when the wind anomalies are present only in the middle or upper stratosphere, the tropospheric response is weak.

      Overall, I found the scientific question in this paper to be well-posed, the experiments to be well-designed, the authors' interpretation of results to be logical and interesting, and the paper to be well-organized and well-written. This manuscript is appropriate for publication in *Weather and Climate Dynamics*, and would make a good contribution to the scientific community. I recommend that this paper be **Accepted with Minor Revisions.** I have a few minor comments for the authors, and a few small suggestions that I think could strengthen this manuscript further, but I do not need to see this manuscript again. My comments and suggestions are presented in line-by-line format below.

**Minor Comments**

1. L17: "**The** troposphere and the stratosphere form a dynamically coupled system."

2. L20: Remove the word "maybe"; you could replace it with "Some of the most prominent stratospheric phenomena…"

3. L31-32: Replace "it can lead to periods with weak and equatorward shifted tropospheric jet stream" with "**a polar vortex break-down c**an lead to periods with **a** weak and equatorward shifted tropospheric jet stream".

4. L66-68:

5. L77: Your left quotation mark around "sub-vortex region" is backwards; if you're using LaTeX for your manuscript, you probably need to use the ` key instead of the ' key.

6. L89: Left quotation mark around "secondary cycles" is backwards; see above.

7. L145: Change "configuration" to "configuration**s**".

8. L146: Change "(slightly)" to "(**a** slightly)".

9. Lines 154-157: When I think about stratosphere-troposphere interactions in the context of SSWs, I often think about the role of planetary-scale waves (e.g., zonal wavenumbers 1-3) rather than the smaller-scale waves in your experiments. Since your results do seem to show

some sensitivity to wavenumber, do you think that larger waves would respond to your stratospheric perturbations in a similar way? Or do you think the behavior would be completely different?

10. Lines 177-178: I found the sentence starting with "However, especially the non-linear decay phase…" a bit confusing. Is this sentence trying to state that wave growth (first few days) is not substantially influenced by the presence of a stratospheric jet? If so, it might help to say that more explicitly. Also, is this result in opposition to some of the results you discuss in your introduction? My understanding is that Wittman et al. (2007) and perhaps Smy and Scott (2009) both saw changes in the growth rates of the baroclinic waves in their experiments, which seems to me to be different to what you saw. This could be a good place to refer back to these earlier works, and perhaps try to explain the discrepancies a bit. I realize you make this comparison a bit later in the paper (around lines 215-220), but I would consider moving this comparison a bit earlier.

11. Line 196, 198: Again, backwards quotation mark around "secondary cycles".

12. Figure 4: Overall, your figures were quite good; however, this figure was a bit confusing for me. We are comparing panels b-d directly to panel a, correct? That is, the shading indicates the difference between the black lines in panels b-d and the black lines in panel a? It might help to say that very explicitly in the caption. I don't think you need to change your figures necessarily (though adding a vertical line at day 6 could be nice but is not necessary), but I would suggest being very direct about what we're looking at, because it took me a couple of minutes to properly orient myself (and you have several other figures that follow this convention, so being painfully obvious about it the first time might be helpful).

    I also would suggest you might want to flip how you describe Figure 4--when I look at panels b-d in Figure 4, the first thing my brain thinks is "equatorward shift". But in the discussion beginning in Line 240, you talk about a poleward shift first, which took me a minute to comprehend. So maybe it would help to mention the equatorward shift without the stratospheric jet BEFORE you mention the poleward shift with the stratospheric jet. Again I don't think you need to change anything about the figure, just the order in which you discuss things.

13. **Discussion in Section 4**: When I was reading this section, I thought back to some of the experiments of Butler et al. (2010). The authors imposed a polar stratospheric cooling anomaly in a simplified model (not winds directly, as you do, but a stronger stratospheric jet was produced in response to the cooling). They tested several different heights of their temperature (and thus, wind) anomalies, and their results seem very complementary to yours--when the temperature anomaly was limited to the middle or upper stratosphere only, the tropospheric response was weaker or non-existent. I'd encourage you to check out their results (centered around their Figure 5), as I think this strengthens some of the points you're making in this section.

    To that end, zonal mean temperature anomalies for your experiments could be interesting, if you have them. My thinking is that it could be nice to more seamlessly link your work to some of the other idealized modeling work that simulates polar stratospheric variability with a temperature anomaly, instead of a wind anomaly (think Figure 4 but for temperature). This is just a thought though--I don't think it would really fit in well in your main manuscript but they could be a good supplemental figure, or even just for your own knowledge.

14. Line 314-15:  Backwards quotation marks around "Set 1" and "Set 2".

15. Line 333:  Remove "**does**" and change "increase" to "increase**s**".

16. Line 364:  Change "worth to point out" to "**worth pointing out**".

17. Lines 365-367:  Butler et al. (2010) (and maybe Polvani and Kushner (2002)) also show increases in eddy momentum fluxes near the tropopause in response to polar stratospheric cooling (and an increase of the winds, similar to your TS), so you could cite that here as well if you'd like.

18. L396:  Change "weather" to "**whether**".

19. Lines 409-418:  Another thing that occurred to me during your discussion of troposphere-only jet variability is that the jet response in many simplified and comprehensive models is dependent on its initial state--that is, jets that start farther equatorward tend to shift more in response to the same perturbation as a jet that starts closer to the pole (e.g., Barnes et al. (2010), Kidston and Gerber (2010)).  Furthermore, the presence or absence of the stratosphere itself can have a large impact on the initial state of the jet (e.g., Wang et al. (2012)).  Since the perturbations that lead to SSW events often come the troposphere, I suppose the point I am trying to make is that the background state can substantially modify the tropospheric jet response to atmospheric perturbations, and the background state of the troposphere can perturb the vortex in the first place.  So at some point when thinking about SSWs it's necessary to zoom out and think about that troposphere-to-stratosphere pathway (not saying you need to change your analysis at all, but I think it is worth mentioning in the discussion at some point).

20. Line 445:  Change "fiction" to "**friction**".

21. Lines 450-455:  Is your explanation for the downward control essentially that changes in lower stratospheric winds (specifically, a weakening of the lower stratospheric winds on the flank of the polar vortex as generally occurs in response to a strong SSW event) drives a negative annular mode-like response in the troposphere? I'd recommend saying that very clearly and explicitly, since this is your last opportunity to concisely summarize your argument.

**References**

1. Barnes, E.A., D.L. Hartmann, D.M.W. Frierson, and J. Kidston (2010):  Effect of latitude on the persistence of eddy-driven jets.  *Geophys. Res. Lett.,* **37**, L11804.

2. Butler, A.H., D.W.J. Thompson, and R. Heike (2010): The steady-state atmospheric circulation response to climate change-like thermal forcings in a simple general circulation model.  *J. Climate*, **23**, 3474-3496.

3. Kidston, J. and E.P. Gerber (2010):  Intermodal variability of the poleward shift of the austral jet stream in the CMIP3 integrations linked to biases in 20th century climatology

4. Wang, S., E. P. Gerber, and L. M. Polvani (2012):  Abrupt circulation responses to tropical upper-tropospheric warming in a relatively simple stratosphere-resolving AGCM.  *J. Climate*, **25**, 4097-4115.

---

## Author Comment (AC1) · 11 Nov 2020

We thank the referee for carefully reading our manuscript, and for their constructive comments. In the following we will respond to the various comments and point out any changes we intend to make to the paper based on them. Note that we have not provided exact manuscript corrections at this point, but we have provided the outline of planned changes. Line numbers and figure references in the reviewer's comments refer to the original manuscript. The reviewer's comments are in black italics; our responses are in blue.

*In this manuscript, the authors use an idealized model to investigate the role of the*

[Figure]

*stratospheric polar vortex on influencing the tropospheric jet. Specifically, they test the sensitivity of the NAM-like equatorward jet shift associated with a weak or reversed stratospheric polar vortex to the height of the vertical winds in the polar jet, and to the magnitude of surface friction. They find that the tropospheric response is sensitive to changes in winds in the lower stratospheric polar vortex, but not to winds in the upper stratosphere. Additionally, they find that surface friction enhances the tropospheric response to stratospheric vortex changes, and that friction acts to bring nearly barotropic anomalies all the way down to the surface.*

Although the findings in general agree with previous studies, the consolidation of the impacts of surface friction and lower-stratospheric anomalies using a coherent model framework produces a compelling standalone study with implications for our dynamical understanding of sudden stratospheric warmings. The manuscript is well-written, well-organized, and the scientific questions well-constructed, and I only have a couple minor comments and questions. Most of my corrections and minor questions can be found in the attached manuscript.

*The main additional question I have is if the authors think this study can provide insight into the observed differences in timing between "Displacement" type sudden stratospheric warming events, and "Split" type sudden stratospheric warming events. Specifically, Splits are observed to show an almost-instantaneous NAM-like response to SSWs, whereas for Displacements, the response tends to occur with a significant lag on the order of weeks. Could this be related to a difference in the vertical structure of the polar vortex in Displacement versus Split events? For example, is it possible that Displacement events show anomalies that start higher up in the stratosphere and work their way downwards, while Split events produce a strong signal in the lower stratosphere almost immediately? I am not sure why or how surface friction could play a role, since I can't imagine any way in which surface friction could depend on the type of vortex breakdown.*

The idea of obtaining insights into differences between displacement and split events is

surely interesting. Smy and Scott (2009) perform life cycle experiments with initial conditions modelling a polar vortex in a split or displacement state based on an imposed PV field and a corresponding PV inversion.

Directly inferring results from the present experiments regarding different SSW states might be a bit speculative and go beyond the scope of this study. However, we think the described set-up and methodology is in general suitable to study such problems and we intend to perform life cycle experiments with more realistic initial conditions (including split and displacement states) at some point. We will add a short note in the discussion about potential extensions of this study.

*Line 155: Should we expect the influence of the stratospheric jet on the tropospheric circulation to peak for tropospheric zonal wave numbers of 6 and 7? Does it have something to do with the relative baroclinic instability for each zonal wave number?*

The strength of the baroclinic life cycle (in our setup) is strongest for wave numbers 6 and 7. When using different wave number perturbations (e.g. 5) the life cycle is in general weaker and therefore the absolute anomaly induced by the inclusion of stratospheric winds is expected to be weaker, too. We will slightly extend the note regarding this wave number sensitivity in our manuscript.

Various typos have been corrected based on the notes within the supplementary material provided by the referee.

---

## Author Comment (AC2) · 11 Nov 2020

We thank the referee for carefully reading our manuscript, and for their constructive comments. In the following we will respond to the various comments and point out any changes we intend to make to the paper based on them. Note that we have not provided exact manuscript corrections at this point, but we have provided the outline of planned changes. Line numbers and figure references in the reviewer's comments refer to the original manuscript. The reviewer's comments are in black italics; our responses are in blue.

[Figure]

*The authors report on a set of baroclinic lifecycle experiments aimed at better understanding the influence of the stratospheric vortex on tropospheric jets. Consistent with previous works, they find that life cycles are substantially modified by presence or lack of a stratospheric vortex. They point out the importance of the winds in the lower stratosphere and identify an amplifying effect of surface friction on the near surface response.*
*The experiments are interesting and potentially quite informative - more idealized experiments such as these are, in my opinion, under-utilized for studying this difficult problem. In particular I found them to be a very clear illustration of the importance of non-linear phase of baroclinic lifecycles in understanding the nature of the tropospheric eddy feedbacks. However, I found some of the discussion to be a bit unsatisfying. I think the most useful suggestion I can give is to strengthen and clarify the comparison of the present results with previous studies on baroclinic lifecycles. Section 1.2 does set up some past results, but it's never made that clear how the present results fit in.*

*For instance: the Wittman papers emphasize changes in linear growth rates, while the present paper emphasizes the non-linear decay phase as the key period of difference, finding (from what I can tell) relatively weak changes in the growth rates. Is this discrepancy consistent with the Smy and Scott emphasis on PV in the sub-vortex region? How much does this have to do with the different basic states? To be clear - overall this is a solid peice of science and should ultimately be worthy of publication. I have some further specific comments and suggestions below; some of which might involve significicant further efforts. I do not require all of these suggestions to be pursued, but I do think the paper would be substantially improved if this main point can be substantially addressed.*

We agree that a clear distinction from previous work (especially previous life cycle work) is very important. We think that the changes in the linear growth phase observed in the

experiments of Wittman et al. and Smy and Scott can be explained by differences in tropopause-level PV gradients and tropospheric winds between their experiments with and without polar vortex, as was noted in Section 3.2. A direct change in tropopause-level PV gradients (as imposed by the initial conditions) leads to modifications of growth phase of the life cycles (e.g., a change in conversion from APE to EKE via the eddy heat flux). In our experiments we tried to specifically minimize this direct stratospheric impact, which can be difficult to distinguish from sensitivities in tropospheric setup. To emphasise this and other major differences in results we will extend the note about the different basic states used in the respective studies and will emphasise at various places in the manuscript that our setup has almost no change in tropopause-level PV gradient in contrast to the setups used in other studies.

*Major comments/suggestions*

*PV structure The PV structure of the initial conditions is emphasized in Fig. 1, but it is not really made reference to later in the interpretation. For instance, how do the perturbations considered in Section 4 modify the PV gradients? One clear difference between T and TS is the presence of narrow region of positive pv gradient near 20 km at 60 degrees in the presence of a stratospheric jet; is it really the winds in the lower stratosphere that are key, or is it the PV gradient structure? Moreover, how is the zonal PV structure changed over the evolution of the lifecycles?*

The main reason for displaying the PV gradients in Fig. 1 was to draw attention the regions of negative gradient that commonly appear in the initial conditions of typical life cycle experiments, but in our experience they are often not reported (although we concluded that those regions have no major effect on the evolution of the life cycle).

The PV distribution (and in particular the PV gradient) are certainly very important in determining the dynamics of the system. However, especially the zonal structure of the response is strongly dictated by the (non-linear) evolution during the decay phase of the life cycle (since the initial state has no zonal structure) and it is not trivial to make

statements about the precise mechanisms or dynamical features of the initial condition setup and processes that lead to the observed NAM signal.

Throughout the manuscript we describe most changes in initial state or life cycle evolution via the wind field (since the system is defined via the wind field), but tried to make it clear that the (structure of the) wind of course couples to various other characteristics of the system, like the temperature and PV fields. Part of the evolution (in particular the zonal structure) is described in Fig. 2 in terms of PV. During our analysis of the system we did not find that using the PV field to describe the final state of the life cycle and the corresponding dependency on stratospheric dynamics to be more useful than the wind field and thus we do not think that adding PV diagnostics to the analysis will improve the manuscript. Further finding crucial differences in PV between the experiments can be difficult since the PV field tends to become rather noisy during the non-linear phase (compared to the wind field), which is even more the case for the PV gradient (as it essentially scales with the curvature of the wind). However, to emphasise the coupling between the various fields and due to general importance of PV (gradient) for the system we will add corresponding notes throughout the manuscript.

*Energetics*

*It seems that one way of interpreting these results, consistent with the discussion of Barnes and Young (1992) in Section 1.2, is that the presence of the stratospheric jet allows for a greater conversion of APE to EKE and ultimately MKE; and that this is further enhanced by including surface friction. If this is a fair characterization of the present results, how does the available potential energy differ in the various cases? Could this result be simply explained by the presence of more APE in the cases with a stratospheric vortex?*

We find that initial state APE is indeed increased in our experiments with stratospheric

jet and ultimately the APE to MKE conversion is enhanced in life cycles with stratospheric jet (consistent with the observed jet shift). However, since this increase in APE corresponds to a change of the temperature structure in the stratosphere it is not obvious that and how the additional stratospheric APE will be converted into tropospheric MKE during the life cycle. Further, we do not find a change in life cycle during the linear growth phase, suggesting the life cycle to be unaffected by the direct increase of initial APE. To address these issues we will extend our discussion of the global energetics and include a note regarding changes in initial state APE and corresponding APE to MKE conversion.

*A related comment: in the presence of surface friction there isn't really final steady state as MKE will continue to dissipate. This is very briefly discussed right at the end of the discussion (l430), but until the reader gets there it's not really clear how the authors defined the final state. Would it be possible to put the figures in the supplementary information on more equal footing by computing the energy lost to surface frection in the runs in Fig. S2, showing instead the net EKE and MKE generated by conservative processes?*

We will add a note just after the introduction of the first experiments including surface friction (Section 3.3) explaining that we define a 'final state' analogous to the case without friction for easy comparison of the two cases. We will further (as also mentioned above) extend our discussion of the global energetics in the discussion section and the supplementary material in order to address the issue of comparability of experiments with and without surface friction.

*Mass fluxes*

*There is some discussion of the 1000 hPa 'Geopotential height' anomalies as a measure of the surface NAM or AO signature of the baroclinic adjustment. One substantial point is that equation (2) on line 270 assumes gradient wind balance holds near surface. This is ok for case with no friction, but not for the case with surface friction, which*

*will modify dominant the force balance. This could be corrected by adding in surface friction on right hand side. But I wonder if it might be more directly useful to consider the evolution of the surface pressure, and the nature of the mass fluxes in these experiments; in particular I wonder if these are substantially modified by the presence of an Ekman layer in the cases with surface friction.*

As the model equations are formulated in absolute pressure surface pressure is not a model variable and corresponding diagnostics are not available. The lack of surface pressure data further does not allow for a direct computation of the geopotential via vertical integration of temperature, which is why we decided to use a definition of geopotential based on the wind field. The geopotential field (on 1000hPa) seems to be a suitable diagnostic to analyse in regards of AO or NAM like phenomena, as is it typically used to define the corresponding indices. We agree that Equation 2 is missing a friction term. However, for friction a time scale of 1 day the missing term $(-a\bar{v}/\tau)$ is typically small compared to the Coriolis term $(-af\bar{u})$. We found a difference in magnitude of the terms of about a factor 100 for the final state of our experiments. We will add a note next to Equation 2 to clarify that we neglect the friction term.

*Specific/Minor Comments:*

*l76: perturbation to what?*

We will add a 'stratospheric PV' before perturbation.

*l188: could this alternatively be explained by greater meridional shear in TS simulation?*

The shear would probably affect wave behaviour via changes in PV gradient, as mentioned. We will add a note to link the PV gradient to the wind structure again as it is probably important to emphasise this connection throughout the paper.

*l213: Are EKE and MKE integrated over depth of troposphere or the depth of atmosphere? I would think that the MKE of polar jet is small but not negligible.*

Energies are integrated over the entire atmosphere (NH only). The difference in MKE

between initial states T and TS is only a small fraction of the MKE difference in their final states. Further, the difference in final state MKE mostly persists when integrating only up to 100hPa, i.e., when we exclude the stratospheric jet from the energy computation. In order to subtract the minor contribution of the stratospheric jet to the MKE we show the difference to the initial state rather than full MKE. We will modify the corresponding parts of the manuscript to clarify the these points.

*l310-317: I found the grouping of these experiments to be initially quite confusing, thinking that the natural ordering would be to group TS>10 with TS<10, and TS>25 with TS<25. I did ultimately figure out that the logic is intended to emphasize the wind anomalies in the lower stratosphere (to be fair, this is stated in the text, but preconceived notions can be hard to shake sometimes).*

*This is ultimately just an issue of presentation. But I did wonder if it would be clearer to readers to start with (what seems to me) the more natural a priori sorting instead of anticipating the results? If not, perhaps naming the sets something more descriptive, such as 'Weak winds/Strong winds' instead of 'Set 1/Set 2'?*

We will change the names of the groups accordingly for more intuitive understanding.

*l325 discussed*

Will be corrected.

*Fig. 7: How does u at 15 km vary over this set of integrations? This might be more illuminating than the mid-stratospheric winds.*

Fig 7 shows final state zonal wind at 10km, which is where we find the strongest response. Winds at 15km change qualitatively the same throughout the simulation as also suggested by the rather barotropic final state winds shown in Fig. 5. Initial state winds at any height change linearly with stratospheric jet magnitude uSmax due to the stratospheric jet being superimposed onto the tropospheric jet. No corresponding changes to the manuscript will be made.

between initial states T and TS is only a small fraction of the MKE difference in their final states. Further, the difference in final state MKE mostly persists when integrating only up to 100hPa, i.e., when we exclude the stratospheric jet from the energy computation. In order to subtract the minor contribution of the stratospheric jet to the MKE we show the difference to the initial state rather than full MKE. We will modify the corresponding parts of the manuscript to clarify the these points.

*l310-317: I found the grouping of these experiments to be initially quite confusing, thinking that the natural ordering would be to group TS>10 with TS<10, and TS>25 with TS<25. I did ultimately figure out that the logic is intended to emphasize the wind anomalies in the lower stratosphere (to be fair, this is stated in the text, but preconceived notions can be hard to shake sometimes).*

*This is ultimately just an issue of presentation. But I did wonder if it would be clearer to readers to start with (what seems to me) the more natural a priori sorting instead of anticipating the results? If not, perhaps naming the sets something more descriptive, such as 'Weak winds/Strong winds' instead of 'Set 1/Set 2'?*

We will change the names of the groups accordingly for more intuitive understanding.

*l325 discussed*

Will be corrected.

*Fig. 7: How does u at 15 km vary over this set of integrations? This might be more illuminating than the mid-stratospheric winds.*

Fig 7 shows final state zonal wind at 10km, which is where we find the strongest response. Winds at 15km change qualitatively the same throughout the simulation as also suggested by the rather barotropic final state winds shown in Fig. 5. Initial state winds at any height change linearly with stratospheric jet magnitude uSmax due to the stratospheric jet being superimposed onto the tropospheric jet. No corresponding changes to the manuscript will be made.

*l413-414: what is this statement based on? explicit calculations or assumption of linearity?*

As mentioned, we performed a set of sensitivity experiments with increased tropospheric jet magnitude (and no stratospheric jet) what showed a similar NAM response as the experiments with stratospheric jet, but the required difference in tropospheric jet magnitude was much larger than the increase induced by inclusion of a stratospheric jet in order to obtain a NAM response of similar magnitude. This suggests that the increase in wind speed at tropopause level alone is not sufficient to explain the observed NAM response in our experiments. We will change the corresponding part to '. . .it was necessary to increase the jet magnitude by order of 10 m/s in these sensitivity experiments. . .' in order to clarify that the statement is based on explicit simulations.

*l423: The additivity of surface winds also is not perfect.*

We will add a corresponding note to the paragraph in order to slightly weaken the statement made.

*l450: This final paragraph also contributed to my feeling unsatisfied by the discussion. What is the explanation for the downward influence suggested by the present results? How can we use these experiments to quantify the eddy feedback? These aren't unreasonable claims, but the authors should explain them more explicitly.*

We agree that a concise, yet complete and clear, description of our main findings and arguments in Section 6 is important and therefore extended the respective last paragraph, as suggested by the referee. .

---

## Author Comment (AC3) · 11 Nov 2020

We thank the referee for carefully reading our manuscript, and for their constructive comments. In the following we will respond to the various comments and point out any changes we intend to make to the paper based on them. Note that we have not provided exact manuscript corrections at this point, but we have provided the outline of planned changes. Line numbers and figure references in the reviewer's comments refer to the original manuscript. The reviewer's comments are in black italics; our responses are in blue.

*This paper posed an interesting scientific question regarding stratosphere-troposphere*

[Figure]

*coupling, and addressed it with well-designed experiments. I think the authors' use of a more simplified atmospheric model was a good choice here, as these simplified simulations are able to capture most of the atmospheric dynamics relevant to their scientific interests. The experiments described in this paper explore the tropospheric response to the presence or absence of a stratospheric jet, in an idealized attempt to compare conditions that typically occur in wintertime (a strong stratospheric jet), and conditions that are common after a sudden stratospheric warming event (a much weaker or absent stratospheric jet). The authors explore a robust parameter space in their experiments, and they analyze a variety of stratospheric jet strengths, several different vertical structures of stratospheric winds, and the effects of surface friction. The authors find that when a stratospheric jet is absent, the tropospheric jet is located equatorward of where it is located in the presence of a stratospheric jet. The presence of surface friction enhances this response, with an even stronger equatorward shift of the tropospheric jet when surface friction is present and a stratospheric jet is absent. Finally, the authors note that the winds in the lower stratosphere are key for influencing the tropospheric response; when the wind anomalies are present only in the middle or upper stratosphere, the tropospheric response is weak.*

*Overall, I found the scientific question in this paper to be well-posed, the experiments to be well-designed, the authors' interpretation of results to be logical and interesting, and the paper to be well-organized and well-written. This manuscript is appropriate for publication in Weather and Climate Dynamics, and would make a good contribution to the scientific community. I recommend that this paper be Accepted with Minor Revisions. I have a few minor comments for the authors, and a few small suggestions that I think could strengthen this manuscript further, but I do not need to see this manuscript again. My comments and suggestions are presented in line-by-line format below.*

*Minor Comments*

*1. L17: " The troposphere and the stratosphere form a dynamically coupled system."*

The text will be corrected accordingly.

*2. L20: Remove the word "maybe"; you could replace it with "Some of the most prominent stratospheric phenomena..."*

The text will be corrected accordingly.

*3. L31-32: Replace "it can lead to periods with weak and equatorward shifted tropospheric jet stream" with " a polar vortex break-down can lead to periods with a weak and equatorward shifted tropospheric jet stream".*

The text will be corrected accordingly.

*4. L66-68:*

*5. L77: Your left quotation mark around "sub-vortex region" is backwards; if you're using LaTeX for your manuscript, you probably need to use the ' key instead of the ' key.*

The text will be corrected accordingly.

*6. L89: Left quotation mark around "secondary cycles" is backwards; see above.*

The text will be corrected accordingly.

*7. L145: Change "configuration" to "configurations ".*

The text will be corrected accordingly.

*8. L146: Change "(slightly)" to "( a slightly)".*

The text will be corrected accordingly.

*9. Lines 154-157: When I think about stratosphere-troposphere interactions in the context of SSWs, I often think about the role of planetary-scale waves (e.g., zonal wavenumbers 1-3) rather than the smaller-scale waves in your experiments. Since your results do seem to show some sensitivity to wavenumber, do you think that larger*

*waves would respond to your stratospheric perturbations in a similar way? Or do you think the behavior would be completely different?*

We agree that planetary waves can play an important role in downward coupling following SSWs. However, our experiments are specifically set up to isolate and study the role of synoptic waves related to baroclinic instability in the troposphere (related to tropospheric eddy feedback). In that sense we are not in a position to say much about the role of planetary waves based on our experiments, which are not primarily controlled by baroclinic instability but rather by forcing due to flow over topography etc. We will include a short remark clarifying this distinction.

*10. Lines 177-178: I found the sentence starting with "However, especially the non-linear decay phase..." a bit confusing. Is this sentence trying to state that wave growth (first few days) is not substantially influenced by the presence of a stratospheric jet? If so, it might help to say that more explicitly. Also, is this result in opposition to some of the results you discuss in your introduction? My understanding is that Wittman et al. (2007) and perhaps Smy and Scott (2009) both saw changes in the growth rates of the baroclinic waves in their experiments, which seems to me to be different to what you saw. This could be a good place to refer back to these earlier works, and perhaps try to explain the discrepancies a bit. I realize you make this comparison a bit later in the paper (around lines 215-220), but I would consider moving this comparison a bit earlier.*

We agree that this would be a useful place to first mention the vanishing influence of stratospheric winds on the linear phase, a corresponding note and reference will be added to the paragraph, but we will leave the main discussion of this issue where it is (within the paragraphs describing Fig. 3).

*11. Line 196, 198: Again, backwards quotation mark around "secondary cycles".*

The text will be corrected accordingly.
*12. Figure 4: Overall, your figures were quite good; however, this figure was a bit confusing for me. We are comparing panels b-d directly to panel a, correct? That is, the shading indicates the difference between the black lines in panels b-d and the black lines in panel a? It might help to say that very explicitly in the caption. I don't think you need to change your figures necessarily (though adding a vertical line at day 6 could be nice but is not necessary), but I would suggest being very direct about what we're looking at, because it took me a couple of minutes to properly orient myself (and you have several other figures that follow this convention, so being painfully obvious about it the first time might be helpful). I also would suggest you might want to flip how you describe Figure 4–when I look at panels b-d in Figure 4, the first thing my brain thinks is "equatorward shift". But in the discussion beginning in Line 240, you talk about a poleward shift first, which took me a minute to comprehend. So maybe it would help to mention the equatorward shift without the stratospheric jet BEFORE you mention the poleward shift with the stratospheric jet. Again I don't think you need to change anything about the figure, just the order in which you discuss things.*

As suggested by the referee we will extend the caption and the describing paragraphs of Fig. 4 to simplify the interpretation of our results. We will further add a visual aid to the figure to help identify day 6.

*13. Discussion in Section 4 : When I was reading this section, I thought back to some of the experiments of Butler et al. (2010). The authors imposed a polar stratospheric cooling anomaly in a simplified model (not winds directly, as you do, but a stronger stratospheric jet was produced in response to the cooling). They tested several different heights of their temperature (and thus, wind) anomalies, and their results seem very complementary to yours–when the temperature anomaly was limited to the middle or upper stratosphere only, the tropospheric response was weaker or non-existent. I'd encourage you to check out their results (centered around their Figure 5), as I think this strengthens some of the points you're making in this section. To that end, zonal mean temperature anomalies for your experiments could be interesting, if you have them. My*

*thinking is that it could be nice to more seamlessly link your work to some of the other idealized modeling work that simulates polar stratospheric variability with a temperature anomaly, instead of a wind anomaly (think Figure 4 but for temperature). This is just a thought though–I don't think it would really fit in well in your main manuscript but they could be a good supplemental figure, or even just for your own knowledge.*

The connection to Butler et al. (2010) is indeed interesting and insightful, further emphasising the range of studies and experimental setups that show a tropospheric jet-shift response to (direct or indirect) wind perturbations in the lower stratosphere. We will add a reference to Butler et al. to the introduction and an additional note to the discussion.

*14. Line 314-15: Backwards quotation marks around "Set 1" and "Set 2".*

The text will be corrected accordingly.

*15. Line 333: Remove " does " and change "increase" to "increases ".*

The text will be corrected accordingly.

*6. Line 364: Change "worth to point out" to " worth pointing out ".*

The text will be corrected accordingly.

*17. Lines 365-367: Butler et al. (2010) (and maybe Polvani and Kushner (2002)) also show increases in eddy momentum fluxes near the tropopause in response to polar stratospheric cooling (and an increase of the winds, similar to your TS), so you could cite that here as well if you'd like.*

We agree that it would be beneficial for the reader to be reminded of the mentioned studies; references will be added to this paragraph as suggested.

*18. L396: Change "weather" to " whether ".*

The text will be corrected accordingly.

*19. Lines 409-418: Another thing that occurred to me during your discussion of troposphere-only jet variability is that the jet response in many simplified and comprehensive models is dependent on its initial state–that is, jets that start farther equatorward tend to shift more in response to the same perturbation as a jet that starts closer to the pole (e.g., Barnes et al. (2010), Kidston and Gerber (2010)). Furthermore, the presence or absence of the stratosphere itself can have a large impact on the initial state of the jet (e.g., Wang et al. (2012)). Since the perturbations that lead to SSW events often come the troposphere, I suppose the point I am trying to make is that the background state can substantially modify the tropospheric jet response to atmospheric perturbations, and the background state of the troposphere can perturb the vortex in the first place. So at some point when thinking about SSWs it's necessary to zoom out and think about that troposphere-to-stratosphere pathway (not saying you need to change your analysis at all, but I think it is worth mentioning in the discussion at some point).*

We agree that it is important to keep in mind that troposphere and stratosphere form a system with coupling in both directions and therefore any climate forcing modifying the tropospheric jet will affect the stratospheric jet, which can then feed-back again onto the tropospheric jet. We will add a note to the discussion section to remind the reader about this fact and include a paragraph on potential future work in the direction of sensitivity of life cycle response to stratospheric conditions to changes in tropospheric initial conditions.

*20. Line 445: Change "fiction" to " friction ".*

The text will be corrected accordingly.

*21. Lines 450-455: Is your explanation for the downward control essentially that changes in lower stratospheric winds (specifically, a weakening of the lower stratospheric winds on the flank of the polar vortex as generally occurs in response to a strong SSW event) drives a negative annular mode-like response in the troposphere?*

*I'd recommend saying that very clearly and explicitly, since this is your last opportunity to concisely summarize your argument.*

We agree that a concise, yet complete and clear, description of our main findings and arguments in Section 6 is important and therefore extended the respective last paragraph, as suggested by the referee.

---

## Author Response (AR1)

We thank the three referees for carefully reading our manuscript, and for their constructive comments. In the following we will respond to the various comments and point out any changes we made to the paper based on them. Line numbers and figure references in the reviewer's comments refer to the original manuscript, while all references in the responses refer to the revised version of the manuscript with tracked changes (from the originally submitted version, shown in red/blue), which can be found at the end of this document. The comments of the referees are in black italics; our responses are in blue.

**Response to Referee 1:**

*The authors report on a set of baroclinic lifecycle experiments aimed at better understanding the influence of the stratospheric vortex on tropospheric jets. Consistent with previous works, they find that life cycles are substantially modified by presence or lack of a stratospheric vortex. They point out the importance of the winds in the lower stratosphere and identify an amplifying effect of surface friction on the near surface response.*
*The experiments are interesting and potentially quite informative - more idealized experiments such as these are, in my opinion, under-utilized for studying this difficult problem. In particular I found them to be a very clear illustration of the importance of non-linear phase of baroclinic lifecycles in understanding the nature of the tropospheric eddy feedbacks. However, I found some of the discussion to be a bit unsatisfying. I think the most useful suggestion I can give is to strengthen and clarify the comparison of the present results with previous studies on baroclinic lifecycles. Section 1.2 does set up some past results, but it's never made that clear how the present results fit in.*

*For instance: the Wittman papers emphasize changes in linear growth rates, while the present paper emphasizes the non-linear decay phase as the key period of difference, finding (from what I can tell) relatively weak changes in the growth rates. Is this discrepancy consistent with the Smy and Scott emphasis on PV in the sub-vortex region? How much does this have to do with the different basic states? To be clear - overall this is a solid peice of science and should ultimately be worthy of publication. I have some further specific comments and suggestions below; some of which might involve significant further efforts. I do not require all of these suggestions to be pursued, but I do think the paper would be substantially improved if this main point can be substantially addressed.*

We agree that a clear distinction from previous work (especially previous life cycle work) is very important. We think that the changes in the linear growth phase observed in the experiments of Wittman et al. and Smy and Scott can be explained by differences in tropopause-level PV gradients and tropospheric winds between their experiments with and without polar vortex, as was already noted in Section 3.2. A direct change in tropopause-level PV gradients (imposed by the initial conditions) leads to modifications of growth phase of the life cycles (e.g., a change in conversion from APE to EKE via the eddy heat flux). In our experiments we tried to specifically minimise this direct stratospheric impact, which can be difficult to distinguish from sensitivities in a purely tropospheric setup. To emphasise this and other major differences in results extend existing notes and added additional notes about the different basic states used in the respective studies at various places in the manuscript (lines 76-78, 95-100, 201-206, 387-394) to further emphasise that our setup has almost no change in tropopause-level PV gradient in contrast to the setups used in other studies and we correspondingly do not observe any changes to the linear growth phase.

**Major comments/suggestions**

**PV structure**

*The PV structure of the initial conditions is emphasized in Fig. 1, but it is not really made reference to later in the interpretation. For instance, how do the perturbations considered in Section 4 modify the PV gradients? One clear difference between T and TS is the presence of narrow region of positive pv gradient near 20 km at 60 degrees in the presence of a stratospheric jet; is it really the winds in the lower stratosphere that are key, or is it the PV gradient structure? Moreover, how is the zonal PV structure changed over the evolution of the lifecycles?*

The main reason for displaying the PV gradients in Fig. 1 was to draw attention to the regions of negative gradient that commonly appear in the initial conditions of typical life cycle experiments, but in our experience they are often not reported (although we concluded that those regions have no major effect on the evolution of the life cycle).

The PV distribution (and in particular the PV gradient) are certainly very important in determining the dynamics of the system. However, especially the zonal structure of the response is strongly dictated by the (non-linear) evolution during the decay phase of the life cycle (since the initial state has no zonal structure) and it is not trivial to make statements about the precise mechanisms or dynamical features of the initial condition setup and processes that lead to the observed NAM signal.

Throughout the manuscript we describe most changes in initial state or life cycle evolution via the wind field (since the system is defined via the wind field), but tried to make it clear that the (structure of the) wind of course couples to various other characteristics of the system, like the temperature and PV fields. Part of the evolution (in particular the zonal structure) is described in Fig. 2 in terms of PV.
During our analysis of the system we did not find that using the PV field to be more useful than the wind field to describe the final state of the life cycle (and the corresponding dependency on stratospheric dynamics) and thus we do not think that adding PV diagnostics to the analysis will improve the manuscript. Further, finding crucial differences in PV between the experiments can be difficult since the PV field tends to become rather noisy during the non-linear phase (compared to the wind field), which is even more the case for the PV gradient (as it essentially scales with the curvature of the wind). However, to emphasise the coupling between PV and the wind field and due to general importance of PV (gradient) for the system we added corresponding note (line 219).

**Energetics**

*It seems that one way of interpreting these results, consistent with the discussion of Barnes and Young (1992) in Section 1.2, is that the presence of the stratospheric jet allows for a greater conversion of APE to EKE and ultimately MKE; and that this is further enhanced by including surface friction. If this is a fair characterization of the present results, how does the available potential energy differ in the various cases? Could this result be simply explained by the presence of more APE in the cases with a stratospheric vortex?*

We find that initial state APE is indeed increased in our experiments with stratospheric jet and ultimately the APE to MKE conversion is enhanced in life cycles with stratospheric jet (consistent with the observed jet shift). However, since this increase in APE corresponds to a change of the temperature structure in the stratosphere it is not obvious that and how the additional stratospheric APE will be converted into tropospheric MKE during the life cycle. Further, we do not find a change in life cycle during the linear growth phase (where APE to EKE conversion is dominant), suggesting the life cycle to be unaffected by the direct increase of initial APE. To address these issues extended our discussion of the global energetics and included a note regarding changes in initial state APE and corresponding APE to MKE conversion (lines 411-419).

*A related comment: in the presence of surface friction there isn't really final steady state as MKE will continue to dissipate. This is very briefly discussed right at the end of the discussion (l430), but until the reader gets there it's not really clear how the authors defined the final state. Would it be possible to put the figures in the supplementary information on more equal footing by computing the energy lost to surface frection in the runs in Fig. S2, showing instead the net EKE and MKE generated by conservative processes?*

We added a note just after the introduction of the first experiments including surface friction (Section 3.3, lines 286-289) explaining that we define a 'final state' analogous to the case without friction for easy comparison of the two cases. We further (as also mentioned above) extended our discussion of the global energetics in the discussion section and the supplementary material in order to address the issue of comparability of experiments with and without surface friction, including added discussions (and Figs. S2 and S3 in the supplement) regarding barotropic energy conversion (lines 260-262, 411-419, 502-504 and supplement).

**Mass fluxes**

*There is some discussion of the 1000 hPa 'Geopotential height' anomalies as a measure of the surface NAM or AO signature of the baroclinic adjustment. One substantial point is that equation (2) on line 270 assumes gradient wind balance holds near surface. This is ok for case with no friction, but not for the case with surface friction, which will modify dominant the force balance. This could be corrected by adding in surface friction on right hand side. But I wonder if it might be more directly useful to consider the evolution of the surface pressure, and the nature of the mass fluxes in these experiments; in particular I wonder if these are substantially modified by the presence of an Ekman layer in the cases with surface friction.*

As the model equations are formulated in absolute pressure surface pressure is not a model variable and corresponding diagnostics are not available. The lack of surface pressure data further does not allow for a direct computation of the geopotential via vertical integration of temperature, which is why we decided to use a definition of geopotential based on the wind field. The geopotential field (on 1000hPa) seems to be a suitable diagnostic to analyse in regards of AO or NAM like phenomena, as is it typically used to define the corresponding indices. We agree that Equation 2 is missing a friction term. However, for a friction time scale of 1 day the missing term ($-a*v\_bar/tau$) is typically small compared to the Coriolis term ($-a*f*u\_bar$). We found a difference in magnitude of the terms of about a factor 100 for the final state of our experiments. We added a note next to Equation 2 to clarify that we neglect the friction term (lines 310-311).

**Specific/Minor Comments:**

*l76: perturbation to what?*

We added a 'stratospheric PV' before perturbation (lines 87-88).

*l188: could this alternatively be explained by greater meridional shear in TS simulation?*

The shear would probably affect wave behaviour via changes in PV gradient, as mentioned. We added a note to link the PV gradient to the wind structure again as it is probably important to emphasise this connection throughout the paper (line 219).

*l213: Are EKE and MKE integrated over depth of troposphere or the depth of atmosphere? I would think that the MKE of polar jet is small but not negligible.*

Energies are integrated over the entire atmosphere (NH only). The difference in MKE between initial states T and TS is only a small fraction of the MKE difference in their final states. Further, the difference in final state MKE mostly persists when integrating only up to

100hPa, i.e., when we exclude the stratospheric jet from the energy computation. In order to subtract the minor contribution of the stratospheric jet to the MKE, we show the difference to the initial state rather than full MKE. We modified the corresponding part of the manuscript to clarify how the energy densities are calculated (line 242).

*l310-317: I found the grouping of these experiments to be initially quite confusing, thinking that the natural ordering would be to group TS>10 with TS<10, and TS>25 with TS<25. I did ultimately figure out that the logic is intended to emphasize the wind anomalies in the lower stratosphere (to be fair, this is stated in the text, but preconceived notions can be hard to shake sometimes).*

*This is ultimately just an issue of presentation. But I did wonder if it would be clearer to readers to start with (what seems to me) the more natural a priori sorting instead of anticipating the results? If not, perhaps naming the sets something more descriptive, such as 'Weak winds/Strong winds' instead of 'Set 1/Set 2'?*

We changex the names of the groups accordingly for more intuitive understanding.

*l325 discussed*

Corrected.

*Fig. 7: How does u at 15 km vary over this set of integrations? This might be more illuminating than the mid-stratospheric winds.*

Fig 7 shows final state zonal wind at 10km, which is where we find the strongest response. Winds at 15km change qualitatively the same throughout the simulation as also suggested by the rather barotropic final state winds shown in Fig. 5. Initial state winds at any height change linearly with stratospheric jet magnitude $u_{Smax}$ due to the stratospheric jet being superimposed onto the tropospheric jet. No corresponding changes to the manuscript were made.

*l413-414: what is this statement based on? explicit calculations or assumption of linearity?*

As mentioned, we performed a set of sensitivity experiments with increased tropospheric jet magnitude (and no stratospheric jet) what showed a similar NAM response as the experiments with stratospheric jet, but the required difference in tropospheric jet magnitude was much larger than the increase induced by inclusion of a stratospheric jet in order to obtain a NAM response of similar magnitude. This suggests that the increase in wind speed at tropopause level alone is not sufficient to explain the observed NAM response in our experiments. We changed the corresponding part to '…it was necessary to increase the jet magnitude by order of 10 m/s in these sensitivity experiments…' in order to clarify that the statement is based on explicit simulations (lines 481-482).

*l423: The additivity of surface winds also is not perfect.*

We added a corresponding note to the paragraph in order to slightly weaken the statement made (line 492).

*l450: This final paragraph also contributed to my feeling unsatisfied by the discussion. What is the explanation for the downward influence suggested by the present results? How can we use these experiments to quantify the eddy feedback? These aren't unreasonable claims, but the authors should explain them more explicitly.*

We agree that a concise, yet complete and clear, description of our main findings and arguments in Section 6 is important and therefore extended the respective last paragraph, as suggested by the referee (lines 533-541).

**Response to Referee 2:**

*In this manuscript, the authors use an idealized model to investigate the role of the stratospheric polar vortex on influencing the tropospheric jet. Specifically, they test the sensitivity of the NAM-like equatorward jet shift associated with a weak or reversed stratospheric polar vortex to the height of the vertical winds in the polar jet, and to the magnitude of surface friction. They find that the tropospheric response is sensitive to changes in winds in the lower stratospheric polar vortex, but not to winds in the upper stratosphere. Additionally, they find that surface friction enhances the tropospheric response to stratospheric vortex changes, and that friction acts to bring nearly barotropic anomalies all the way down to the surface.*

*Although the findings in general agree with previous studies, the consolidation of the impacts of surface friction and lower-stratospheric anomalies using a coherent model framework produces a compelling standalone study with implications for our dynamical understanding of sudden stratospheric warmings. The manuscript is well-written, well-organized, and the scientific questions well-constructed, and I only have a couple minor comments and questions. Most of my corrections and minor questions can be found in the attached manuscript.*

*The main additional question I have is if the authors think this study can provide insight into the observed differences in timing between "Displacement" type sudden stratospheric warming events, and "Split" type sudden stratospheric warming events. Specifically, Splits are observed to show an almost-instantaneous NAM-like response to SSWs, whereas for Displacements, the response tends to occur with a significant lag on the order of weeks. Could this be related to a difference in the vertical structure of the polar vortex in Displacement versus Split events? For example, is it possible that Displacement events show anomalies that start higher up in the stratosphere and work their way downwards, while Split events produce a strong signal in the lower stratosphere almost immediately? I am not sure why or how surface friction could play a role, since I can't imagine any way in which surface friction could depend on the type of vortex breakdown.*

The idea of obtaining insights into differences between displacement and split events is surely interesting. Smy and Scott (2009) perform life cycle experiments with initial conditions modelling a polar vortex in a split or displacement state based on an imposed PV field and a corresponding PV inversion.
Directly inferring results from the present experiments regarding different SSW states might be a bit speculative and go beyond the scope of this study. However, we think the described set-up and methodology is in general suitable to study such problems and we intend to perform life cycle experiments with more realistic initial conditions (including split and displacement states) at some point. We added a paragraph in the discussion about potential extensions of this study (lines 505-515).

*Line 155: Should we expect the influence of the stratospheric jet on the tropospheric circulation to peak for tropospheric zonal wave numbers of 6 and 7? Does it have something to do with the relative baroclinic instability for each zonal wave number?*

The strength of the baroclinic life cycle (in our setup) is strongest for wave numbers 6 and 7. When using different wave number perturbations (e.g. 5) the life cycle is in general weaker and therefore the absolute anomaly induced by the inclusion of stratospheric winds is expected to be weaker, too. We extended the note regarding this wave number sensitivity in our manuscript (lines 178-181).

Various typos have been corrected based on the notes within the supplementary material provided by the referee.

**Response to Referee 3:**

*This paper posed an interesting scientific question regarding stratosphere-troposphere coupling, and addressed it with well-designed experiments. I think the authors' use of a more simplified atmospheric model was a good choice here, as these simplified simulations are able to capture most of the atmospheric dynamics relevant to their scientific interests. The experiments described in this paper explore the tropospheric response to the presence or absence of a stratospheric jet, in an idealized attempt to compare conditions that typically occur in wintertime (a strong stratospheric jet), and conditions that are common after a sudden stratospheric warming event (a much weaker or absent stratospheric jet). The authors explore a robust parameter space in their experiments, and they analyze a variety of stratospheric jet strengths, several different vertical structures of stratospheric winds, and the effects of surface friction. The authors find that when a stratospheric jet is absent, the tropospheric jet is located equatorward of where it is located in the presence of a stratospheric jet. The presence of surface friction enhances this response, with an even stronger equatorward shift of the tropospheric jet when surface friction is present and a stratospheric jet is absent. Finally, the authors note that the winds in the lower stratosphere are key for influencing the tropospheric response; when the wind anomalies are present only in the middle or upper stratosphere, the tropospheric response is weak.*

*Overall, I found the scientific question in this paper to be well-posed, the experiments to be well-designed, the authors' interpretation of results to be logical and interesting, and the paper to be well-organized and well-written. This manuscript is appropriate for publication in Weather and Climate Dynamics, and would make a good contribution to the scientific community. I recommend that this paper be Accepted with Minor Revisions. I have a few minor comments for the authors, and a few small suggestions that I think could strengthen this manuscript further, but I do not need to see this manuscript again. My comments and suggestions are presented in line-by-line format below.*

***Minor Comments***
*1. L17: " The troposphere and the stratosphere form a dynamically coupled system."*
      The text was corrected accordingly.

*2. L20: Remove the word "maybe"; you could replace it with "Some of the most prominent stratospheric phenomena…"*
      The text was corrected accordingly.

*3. L31-32: Replace "it can lead to periods with weak and equatorward shifted tropospheric jet stream" with " a polar vortex break-down can lead to periods with a weak and equatorward shifted tropospheric jet stream".*
      The text was corrected accordingly.

*4. L66-68:*

*5. L77: Your left quotation mark around "sub-vortex region" is backwards; if you're using LaTeX for your manuscript, you probably need to use the ` key instead of the ' key.*
      The text was corrected accordingly.

*6. L89: Left quotation mark around "secondary cycles" is backwards; see above.*
      The text was corrected accordingly.

*7. L145: Change "configuration" to "configurations ".*

*The text was corrected accordingly.*

*8. L146: Change "(slightly)" to "( a slightly)".*
*The text was corrected accordingly.*

*9. Lines 154-157: When I think about stratosphere-troposphere interactions in the context of SSWs, I often think about the role of planetary-scale waves (e.g., zonal wavenumbers 1-3) rather than the smaller-scale waves in your experiments. Since your results do seem to show some sensitivity to wavenumber, do you think that larger waves would respond to your stratospheric perturbations in a similar way? Or do you think the behavior would be completely different?*

We agree that planetary waves can play an important role in downward coupling following SSWs. However, our experiments are specifically set up to isolate and study the role of synoptic waves related to baroclinic instability in the troposphere (related to tropospheric eddy feedback). In that sense we are not in a position to say much about the role of planetary waves based on our experiments, which are not primarily controlled by baroclinic instability but rather by forcing due to flow over topography etc. We included a short remark clarifying this distinction (lines 178-181).

*10. Lines 177-178: I found the sentence starting with "However, especially the non-linear decay phase…" a bit confusing. Is this sentence trying to state that wave growth (first few days) is not substantially influenced by the presence of a stratospheric jet? If so, it might help to say that more explicitly. Also, is this result in opposition to some of the results you discuss in your introduction? My understanding is that Wittman et al. (2007) and perhaps Smy and Scott (2009) both saw changes in the growth rates of the baroclinic waves in their experiments, which seems to me to be different to what you saw. This could be a good place to refer back to these earlier works, and perhaps try to explain the discrepancies a bit. I realize you make this comparison a bit later in the paper (around lines 215-220), but I would consider moving this comparison a bit earlier.*

We agree that this would be a useful place to first mention the vanishing influence of stratospheric winds on the linear phase, a corresponding note and reference were added to the paragraph (lines 201-206), but we left the main discussion of this issue where it is (within the paragraphs describing Fig. 3). We further added notes throughout the manuscript emphasising the distinction between our findings regarding the linear growth phase and the findings of other authors (95-100, 201-206, 387-394).

*11. Line 196, 198: Again, backwards quotation mark around "secondary cycles".*
*The text was corrected accordingly.*

*12. Figure 4: Overall, your figures were quite good; however, this figure was a bit confusing for me. We are comparing panels b-d directly to panel a, correct? That is, the shading indicates the difference between the black lines in panels b-d and the black lines in panel a? It might help to say that very explicitly in the caption. I don't think you need to change your figures necessarily (though adding a vertical line at day 6 could be nice but is not necessary), but I would suggest being very direct about what we're looking at, because it took me a couple of minutes to properly orient myself (and you have several other figures that follow this convention, so being painfully obvious about it the first time might be helpful). I also would suggest you might want to flip how you describe Figure 4--when I look at panels b-d in Figure 4, the first thing my brain thinks is "equatorward shift". But in the discussion beginning in Line 240, you talk about a poleward shift first, which took me a minute to comprehend. So maybe it would help to mention the equatorward shift without the stratospheric jet BEFORE you mention the poleward shift with the stratospheric jet. Again I don't think you need to change anything about the figure, just the order in which you discuss things.*

As suggested by the referee we extended the caption and the describing paragraphs of Fig. 4 to simplify the interpretation of our results (lines 270-273). We further added a visual aid to the figure to help identify day 6.

*13. Discussion in Section 4 : When I was reading this section, I thought back to some of the experiments of Butler et al. (2010). The authors imposed a polar stratospheric cooling anomaly in a simplified model (not winds directly, as you do, but a stronger stratospheric jet was produced in response to the cooling). They tested several different heights of their temperature (and thus, wind) anomalies, and their results seem very complementary to yours--when the temperature anomaly was limited to the middle or upper stratosphere only, the tropospheric response was weaker or non-existent. I'd encourage you to check out their results (centered around their Figure 5), as I think this strengthens some of the points you're making in this section. To that end, zonal mean temperature anomalies for your experiments could be interesting, if you have them. My thinking is that it could be nice to more seamlessly link your work to some of the other idealized modeling work that simulates polar stratospheric variability with a temperature anomaly, instead of a wind anomaly (think Figure 4 but for temperature). This is just a thought though--I don't think it would really fit in well in your main manuscript but they could be a good supplemental figure, or even just for your own knowledge.*

The connection to Butler et al. (2010) is indeed interesting and insightful, further emphasising the range of studies and experimental setups that show a tropospheric jet-shift response to (direct or indirect) wind perturbations in the lower stratosphere. We added references to Butler et al. to the introduction and an additional note to the discussion (lines 47-51, 429-431, 468-469).

*14. Line 314-15: Backwards quotation marks around "Set 1" and "Set 2".*

The text was corrected accordingly.

*15. Line 333: Remove " does " and change "increase" to "increases ".*

The text was corrected accordingly.

*16. Line 364: Change "worth to point out" to " worth pointing out ".*

The text will be corrected accordingly.

*17. Lines 365-367: Butler et al. (2010) (and maybe Polvani and Kushner (2002)) also show increases in eddy momentum fluxes near the tropopause in response to polar stratospheric cooling (and an increase of the winds, similar to your TS), so you could cite that here as well if you'd like.*

We agree that it would be beneficial for the reader to be reminded of the mentioned studies; references were added to this paragraph as suggested (lines 429-431).

*18. L396: Change "weather" to " whether ".*

The text was corrected accordingly.

*19. Lines 409-418: Another thing that occurred to me during your discussion of troposphere-only jet variability is that the jet response in many simplified and comprehensive models is dependent on its initial state--that is, jets that start farther equatorward tend to shift more in response to the same perturbation as a jet that starts closer to the pole (e.g., Barnes et al. (2010), Kidston and Gerber (2010)). Furthermore, the presence or absence of the stratosphere itself can have a large impact on the initial state of the jet (e.g., Wang et al. (2012)). Since the perturbations that lead to SSW events often come the troposphere, I suppose the point I am trying to make is that the background state can substantially modify the tropospheric jet response to atmospheric perturbations, and the background state of the troposphere can perturb the vortex in the first place. So at some point when thinking about SSWs it's necessary*

*to zoom out and think about that troposphere-to-stratosphere pathway (not saying you need to change your analysis at all, but I think it is worth mentioning in the discussion at some point).*

We agree that it is important to keep in mind that troposphere and stratosphere form a system with coupling in both directions and therefore any climate forcing modifying the tropospheric jet will also affect the stratospheric jet, which can then feed-back again onto the tropospheric jet. We added a note to the discussion section to remind the reader about this fact (lines 424-425).

*20. Line 445: Change "fiction" to " friction ".*

The text will be corrected accordingly.

*21. Lines 450-455: Is your explanation for the downward control essentially that changes in lower stratospheric winds (specifically, a weakening of the lower stratospheric winds on the flank of the polar vortex as generally occurs in response to a strong SSW event) drives a negative annular mode-like response in the troposphere? I'd recommend saying that very clearly and explicitly, since this is your last opportunity to concisely summarize your argument.*

We agree that a concise, yet complete and clear, description of our main findings and arguments in Section 6 is important and therefore extended the respective last paragraph, as suggested by the referee (lines 533-541).

[revised manuscript text omitted]

40    influence on the tropospheric circulation. However, no single fully conclusive mechanism has been  identified yet. Note that, in addition, the tropospheric response to SSWs might also be caused by a combination of different coupling processes. One of these potential coupling processes is given by tropospheric eddy feedback as a response to the induced stratospheric anomalies. Domeisen et al. (2013) have shown in idealised model runs that tropospheric  eddy feedback is essential to obtain a robust negative NAM signal following a SSW. Hitchcock and Simpson (2014) also found tropospheric synoptic-scale

45    eddy feedback to play a significant role in creating a NAM-like surface response. They further concluded that the most relevant aspect of the stratospheric variability does not seem to be the wind reversal in the mid-stratosphere, but the persistent wind anomalies in the lowermost stratosphere. Butler et al. (2010) performed a series of steadily forced idealised model experiments with imposed cooling either in the entire polar stratosphere or confined to the middle and upper polar stratosphere (mimicking, e.g., ozone-hole induced climate change), directly causing a consistent wind anomaly in the respective region. They found the

50    troposphere to respond to the imposed stratospheric anomalies in a NAM-like fashion if the imposed anomalies reach into the lower stratosphere, but the tropospheric response to be weak if the anomalies are confined to the upper stratosphere. Karpechko et al. (2017) showed that in both, model runs and reanalysis data, SSWs which produce strong and long-lasting anomalies in the lowermost stratosphere have an increased likelihood for a tropospheric impact compared to SSWs with weak anomalies in the lowermost stratosphere.

55    ## 1.2 Previous baroclinic life cycle work relevant for this study

A simple, yet fundamental, way to investigate the role of synoptic scale eddies in the dynamical coupling between stratosphere and troposphere is through (idealised) baroclinic life cycle experiments, an initial value problem starting from an imposed baroclinically unstable tropospheric jet. During the subsequent break-down of the imposed jet a baroclinic wave can be observed to

develop, grow and eventually decay, leaving the system in a state with a more barotropic, strengthened and poleward shifted jet

60    compared to the initial conditions (see, e.g., Simmons and Hoskins, 1978; Thorncroft et al., 1993). Such life cycle experiments have previously been used to study the influence of stratospheric winds onto the evolution of tropospheric baroclinic eddies.

Wittman et al. (2004) performed idealised life cycle experiments using initial conditions that either do or do not include winds in the stratosphere, representing situations with an intact or a broken-down polar vortex. They found that if the system includes a polar vortex the evolution of the life cycle is strongly modified and when the polar vortex is removed the system

65    exhibits a (weak) dipole structure in the surface geopotential height field, similar to the surface NAM response observed after SSWs, which corresponds to an equatorward shift of the tropospheric jet. They further note that this surface signal is weak if the polar vortex is rather confined to the stratosphere, but gets strongly enhanced if the polar vortex reaches deep into the troposphere.

In a following study Wittman et al. (2007) investigated the role of stratospheric vertical shear onto the evolution of baroclinic

70    life cycles. They used three different setups in which the winds of the tropospheric jet either decreased, stayed constant or (further) increased above the jet core. For the three situations they found pronounced differences in the evolution of the life cycle, including substantial changes in the growth rate of the baroclinic waves and the qualitative characteristics of the wave growth and decay phases. It should be noted, that the initial conditions used by Wittman et al. (2007) were mostly motivated to resemble a setup of the Eady model for baroclinic instability, rather than realistic atmospheric conditions. The corresponding

75    change of stratospheric shear induces strong changes in the vertical curvature of zonal wind at tropopause level, and thus strong changes in the meridional gradient of potential vorticity (PV) in that region, which are known to have a strong impact on the evolution of baroclinic waves in the troposphere. In the present study we specifically design initial conditions that do not substantially modify tropopause-level PV gradients to minimise their direct impact on the development of baroclinic instability.

[revised manuscript text omitted]

205    (e.g., Wittman et al. (2007); Kunz et al. (2009); Smy and Scott (2009)). We will discuss this apparent contradiction 
[revised manuscript text omitted]

changes in stratospheric conditions. These findings seem to contradict previously reported results (e.g.,

390  Wittman et al. (2007); Kunz et al. (2009); Smy and Scott (2009)). A likely explanation for this discrepancy is that the linear growth phase is highly sensitive to tropopause-level PV gradients of the initial conditions, which had been altered due to changes in the stratosphere in these previous studies. The structure of the PV gradient in our experiments, on the other hand, is essentially not altered by the inclusion of stratospheric winds and the linear growth of baroclinic waves (driven by tropospheric heat fluxes) is therefore unchanged. However, we found the presence of a stratospheric jet to substantially alter the non-linear

395  decay phase of  the life cycle.  Notably it is during this decay phase where the eddy momentum flux acts to convert eddy kinetic to mean kinetic energy while at the same time driving a meridional shift of the tropospheric jet. Given

that the momentum flux is proportional to the equatorward wave activity flux our findings are consistent with a stratospheric influence via modifications to meridional wave propagation near the tropopause.

Figures 2 and 3 showed changes in the secondary cycles occurring during the non-linear decay stage, including changes in number, strength, duration, timing and apparent type (or flavour) of these secondary cycles. The different types of baroclinic wave breaking (LC1 and LC2) have been linked to different weather regimes, and thus corresponding transitions within a life cycle can potentially have a large impact on surface weather (e.g., Michel and Rivière, 2011). As discussed, the baroclinic decay phase of experiment TS shows characteristics of both, LC1 and LC2 flavour, or equivalently cyclonic and anticyclonic wave breaking, while experiment T shows more LC1 characteristics. However, the general behaviour of the TS life cycle (e.g., in terms of its final state) still follows primarily the (anticyclonic) LC1 type, and it only seems to experience individual, transient (cyclonic) LC2 wave-breaking events. The importance of these transient LC2 events for the overall meteorological regime is presently not clear. Note that other authors have previously also reported about transitions between LC1 and LC2 wave breaking states based on stratospheric conditions (e.g., Kunz et al., 2009), but mostly in terms of the entire life cycle, rather than in terms of more transient events.

The

Further note that the introduction of a stratospheric jet not only modifies the details of the wave breaking during the decay phase, but also the (steadyquasi-steady) final state of the life cycle. In particular, do generally elevated values of EKE during the decay phase and increased MKE during the final state suggest an enhanced barotropic EKE to MKE conversion (as also seen in Fig. S2 in the supplementary material). The inclusion of a stratospheric jet to a basic state also formally corresponds to an increase in MPE (mean potential energy, also referred to as available potential energy or APE), where the latter in principle forms the main energy source for the life cycle and is, via EKE, ultimately converted into MKE. However, since this increase in MPE is primarily associated with stratospheric temperature structure, it is unlikely to contribute to the predominantly tropospheric energy conversions during the life cycle.

Consistent with the change in final state MKE we observed a relative equatorward shift of the tropospheric jet in the final state when removing the stratospheric jet from the initial conditions of the system, as can be seen in Figures 5 and 6. This relative jet shift is analogous to the NAM-like signature that has been observed after SSW events. To what extent the observed NAM response to SSWs is similarly influenced by tropospheric eddy feedbacks, as suggested by our results, remains to be quantified further.

It is worth to It also is important to remember that the coupling of troposphere and stratosphere works in both directions and the coupled system will generally react as a whole to any (tropospheric or stratospheric) forcing.

We further want to point out that the relative meridional shift between the final jet in the experiments T and TS results from differences in the meridional eddy momentum transport during the life cycle (not shown). The increased momentum fluxes in experiment TS, compared to T, can be related to increased wave activity around tropopause level, which is consistent with the increase in EKE shown in Figure 3. Similar changes in eddy momentum transport as a result to (in particular lower-stratospheric) climate anomalies have been observed previously in idealised general circulation model experiments by various authors (e.g., Polvani and Kushner, 2002; Butler et al., 2010).

Figure 5 further shows that the surface signal of the NAM-response in the final state, given by the zonal mean zonal wind difference between experiments T and TS, is enhanced when the system is subject to surface friction. The effect of surface friction to increase the surface wind signal of the NAM-response might seem counter-intuitive. However, as we already pointed out in Subsection 1.3, surface friction can provide a way for tropopause level eddy momentum fluxes to couple to the surface winds. The modification of the baroclinic eddy field by the presence of a stratospheric jet can therefore project more strongly onto the surface winds and produce a stronger surface signal. The evolution equation of the vertically averaged zonal mean zonal wind (Equation 1) is often used to argue that on long time scales (where $\partial_t[\bar{u}] \approx 0$) the eddy flux convergence has to be balanced by the (dissipation of) surface winds. In our (transient) life cycle experiments we cannot neglect the wind tendency term and the main balance is given by $\partial_t[\bar{u}] \approx -\partial_y\left[\overline{u'v'}\right]$. However, the dissipation term $\bar{u}_{sfc}/\tau$ provides an important contribution to the equation and strongly modifies the acceleration of the jet, as is already suggested by the factor 2 difference of the final jet magnitude in the cases with and without surface friction (see Figure 5).

When interpreting Equation 1 one also has to keep in mind that it describes the evolution of the full (vertically and zonally averaged) wind field, whereas we are mostly interested in the enhanced surface signal of the difference in wind field between experiments TS and T (see Figure 5), i.e., the NAM-like shift signal. The line of argumentation, however, is analogous. The introduction (or removal) of a stratospheric jet influences the evolution of baroclinic eddies at tropopause level. Following Equation 1, the corresponding changes in eddy momentum flux then induce changes in the wind tendency (which will tend to be close to the level of the eddy flux, primarily near the tropopause), but also couple directly to the surface winds.

The enhancement of the surface signal by surface friction can potentially be understood via the following mechanism: the decay stage of the life cycle is characterised by a barotropisation of the tropospheric jet and thus a reduction of vertical shear and a strengthening of surface winds. The surface friction, on the other hand, tends to increase the vertical wind shear and  therefore act as a source of baroclinicity. This increase in baroclinicity then leads to  an enhanced barotropisation of the jet (similar to what was observed by Barnes and Young, 1992) and a correspondingly enhanced downward propagation of the jet-shift signal. The idea of an enhanced baroclinisation/barotropisation is consistent with the formation of additional life cycles which we observed during the late stages of the life cycle  in experiments with surface friction (seen roughly between days 15 and 25 in Fig. S3 in the supplementary material).

Figure 7 indicated that the strength of the NAM-response following the removal of the stratospheric jet depends non-linearly on the magnitude of the stratospheric jet. In particular, the signal seems to saturate when the stratospheric jet magnitude exceeds a certain value and stronger jets do not lead to a stronger NAM-signal any more. This behaviour suggests that an anomalously strong polar vortex does not necessarily lead to anomalously positive NAM-signals. Similarly Figure 7 indicates that a reversal of the stratospheric jet (with $u_{Smax} < 0$) does not lead to a negative NAM-response with respect to experiment T, which suggests that in terms of NAM-response it is not important  whether a SSW leads to slightly or strongly reversed winds of the polar vortex. However, the setup of baroclinic life cycle experiments does, of course, not capture the dynamics around SSWs in their entire complexity and these results do not necessarily carry over to the real atmosphere.

In Section 4 we showed that the NAM-response observed in the final state of our life cycle experiments is mostly caused by the change in wind structure in the lower stratosphere when including the stratospheric jet, rather than wind anomalies in the middle and upper stratosphere ( Figure 10).  The sensitivity of the eddy feedback to wind anomalies in the lower stratosphere is consistent with results previously reported by various authors (e.g., Butler et al., 2010).

470  It should be noted that changing the wind structure in the lower stratosphere does also introduce changes in various other characteristics of the corresponding initial conditions, like the height of maximum wind speed, the vertical wind shear in the upper troposphere (roughly up to 10km) and the magnitude of the tropospheric jet (especially obvious for profiles T and TS in Figure 9). However, these three characteristics are intrinsically not completely independent and can potentially all affect the evolution of the life cycle. This can be seen, e.g., since the vertical wind shear is (via thermal wind balance) related to

475  the horizontal temperature gradient, which drives the growth of baroclinic waves and can, among other things, modify their (linear) growth rate (although note that the near surface shear is almost identical in the different experiments).

We performed a set of sensitivity experiments (not shown) with tropospheric jet only and varying tropospheric jet magnitude (and therefore increased vertical shear in the troposphere). We found that an increase in tropospheric jet strength also leads to a increased poleward shift during the life cycle (i.e., an equatorward shift of the jet in the final state of experiment T relative to

480  a case with stronger tropospheric jet), similar to the shift observed in Figure 5a. In order to achieve a jet shift signal of similar magnitude as the one shown in Figure 5, however, it was necessary to increase the jet magnitude by order of 10 m/s in these sensitivity experiments (the difference in tropospheric jet magnitude between experiments T and TS is only of the order of 1 m/s.), indicating that other characteristics of the initial state need to contribute and the observed jet shift cannot purely be a result of a strengthened tropospheric jet. The inclusion of the stratospheric jet does to some extend project onto the mentioned

485  characteristics (e.g., height of the jet core and tropospheric shear) of the total zonal wind profile and the resulting jet shift can potentially be interpreted as the result of a combination of factors.

Figure 11 further suggested that we essentially recover the surface geopotential height signal of experiment TS (with full stratospheric jet), when adding the corresponding signals of experiments $T_{<10}$ and $T_{>10}$. Such additivity of responses might be another indication that the stratospheric jet projects onto various other structures and characteristics (e.g., tropospheric

490  shear and jet core height) and the corresponding jet shift response forms as the result of a combination of responses to those modifications. However, while the anomalies of the respective experiments seem to be additive when it comes to the surface geopotential height (although not perfect), the middle tropospheric jet shift response in Figure 10 does not appear to follow the same additive behaviour.

As discussed, Figure 10  shows the NAM-like jet shift signature of the life cycle due to the inclusion of a strato-

495  spheric jet to be mainly caused by the corresponding change in winds in the lower stratosphere, rather than the winds in the middle and upper stratosphere, where the stratospheric jet itself is strongest. A similar conclusion can be drawn from the energetics of the system (provided as supplementary material), which shows a consistent increase in MKE of the final state for the experiments  with strong winds in the lower stratosphere (as defined in Section 4), compared to the experiments  with weak winds, in a system that does not include surface friction. As also explained in Section 3, this increase in

500  MKE is caused by the relative meridional shift of the final tropospheric jet. Note that if the system includes surface friction the

constant dissipation of winds leads to a gradual and flow dependent decrease of MKE, which makes the interpretation of the energetics in terms of a final state difficult. However, we find basic states which include a stratospheric jet to be associated with an enhanced barotropic energy conversion of EKE to MKE during the life cycle in both, systems that do and do not include surface friction (see Fig. S2 in the supplement).

While the present study discusses the sensitivity of the tropospheric jet shift to the presence of a stratospheric jet during baroclinic life cycles in some detail, various questions remain open and provide potential for future work. It might be possible to use the idealised life cycle set-up discussed above to gain insights into the distinction between the responses to different types of SSWs. For example, sudden stratospheric warming events characterised as split or displacement have been found to be associated with different lower-stratospheric wind anomalies, while the question of differences in their tropospheric response has not been fully answered (see Charlton and Polvani, 2007; Mitchell et al., 2011; Maycock and Hitchcock, 2015). Smy and Scott (2009) studied the distinction of split and displacement events in idealised life cycle experiments and found strong differences in tropospheric response. However, it should be noted that their experiments are initialised with a basic state constructed via the inversion of an imposed PV field which will inevitably have an influence on the tropospheric initial state and thus directly affect the evolution of the life cycle. The distinct stratospheric influences via direct remote PV signatures or indirect influences on non-linear baroclinic eddy dynamics remains to be investigated further.

**6 Summary and conclusions**

In this paper we discussed changes in the evolution of idealised baroclinic life cycles induced by the presence of a stratospheric jet. Particular focus was given on a jet shift signal in the zonal wind anomaly of the final state of the life cycle, similar to the signature of negative (surface) anomalies of the northern annular mode (NAM) often observed after sudden stratospheric warming (SSW) events.

We found that the final state of the life cycle is associated with increased zonal mean kinetic energy when a stratospheric jet is included in the system, roughly representing the polar vortex of typical winter-time conditions, compared to the typical life cycle setup including only a tropospheric jet, roughly representing post-SSW conditions. This increase in mean kinetic energy corresponds to a negative NAM signal in the final state zonal wind, i.e., a relative equatorward shift of the tropospheric jet in the case with tropospheric jet only compared to the case with tropospheric and stratospheric jet. The negative NAM signal is the result of a reduced poleward shift over the course of the life cycle induced by a reduction in eddy momentum transport at tropopause level.

The corresponding NAM-like jet-shift response has an increased surface signal if the system includes surface friction, which might seem counter-intuitive, but is consistent with the idea of an increased coupling of surface winds to the eddy momentum transport at tropopause level due to the friction.

We further showed that the system is mainly sensitive to changes of the wind structure in the lower stratosphere (heights between 10 km and 25 km), rather than to zonal wind anomalies in the middle and upper stratosphere.

The findings of this paper provide further evidence for the role of tropospheric eddy feedbacks in shaping the tropospheric response to stratospheric events. In particular, they help to explain the observed negative surface NAM signal following SSWs. The  simplified nature of the idealised life cycle setup allows for a clean separation of tropospheric eddy feedbacks in the surface response to different stratospheric conditions, highlighting the role of tropopause-level momentum fluxes in the non-linear phase of the life cycle. It furthermore offers quantitative insights into the role of surface friction in modulating the surface response to stratospheric events in a simplified setting.

**Appendix A:  Appendix: Construction of initial state**

[revised manuscript text omitted]

Maycock, A. C. and Hitchcock, P.: Do split and displacement sudden stratospheric warmings have different annular mode signatures?, Geophysical Research Letters, 42, 10–943, 2015.

Michel, C. and Rivière, G.: The link between Rossby wave breakings and weather regime transitions, Journal of the Atmospheric Sciences, 68, 1730–1748, 2011.

Mitchell, D. M., Charlton-Perez, A. J., and Gray, L. J.: Characterizing the variability and extremes of the stratospheric polar vortices using 2D moment analysis, Journal of the atmospheric sciences, 68, 1194–1213, 2011.

Polvani, L. M. and Esler, J.: Transport and mixing of chemical air masses in idealized baroclinic life cycles, Journal of Geophysical Research: Atmospheres, 112, 2007.

Polvani, L. M. and Kushner, P. J.: Tropospheric response to stratospheric perturbations in a relatively simple general circulation model, Geophysical Research Letters, 29, 18–1, 2002.

Polvani, L. M., Scott, R., and Thomas, S.: Numerically converged solutions of the global primitive equations for testing the dynamical core of atmospheric GCMs, Monthly weather review, 132, 2539–2552, 2004.

Rivier, L., Loft, R., and Polvani, L. M.: An efficient spectral dynamical core for distributed memory computers, Monthly weather review, 130, 1384–1396, 2002.

Simmons, A. J. and Hoskins, B. J.: The life cycles of some nonlinear baroclinic waves, Journal of the Atmospheric Sciences, 35, 414–432, 1978.

Smy, L. and Scott, R.: The influence of stratospheric potential vorticity on baroclinic instability, Quarterly Journal of the Royal Meteorological Society: A journal of the atmospheric sciences, applied meteorology and physical oceanography, 135, 1673–1683, 2009.

Thompson, D. W. and Wallace, J. M.: The Arctic Oscillation signature in the wintertime geopotential height and temperature fields, Geophysical research letters, 25, 1297–1300, 1998.

Thompson, D. W. and Wallace, J. M.: Regional climate impacts of the Northern Hemisphere annular mode, Science, 293, 85–89, 2001.

Thorncroft, C., Hoskins, B., and McIntyre, M.: Two paradigms of baroclinic-wave life-cycle behaviour, Quarterly Journal of the Royal Meteorological Society, 119, 17–55, 1993.

Vallis, G. K.: Atmospheric and Oceanic Fluid Dynamics: Fundamentals and Large-Scale Circulation, Cambridge University Press, 2 edn., https://doi.org/10.1017/9781107588417, 2017.

Wittman, M. A., Polvani, L. M., Scott, R. K., and Charlton, A. J.: Stratospheric influence on baroclinic lifecycles and its connection to the Arctic Oscillation, Geophysical research letters, 31, 2004.

Wittman, M. A., Charlton, A. J., and Polvani, L. M.: The effect of lower stratospheric shear on baroclinic instability, Journal of the atmospheric sciences, 64, 479–496, 2007.

---

## Author Response (AR2)

We thank the referees for carefully reading our manuscript again, and for their additional constructive comments, to which we will respond in the following and point out any changes made based on them. Line numbers and figure references in the comments refer to the first version of the revised manuscript. The reviewer's comments are in black italics; our responses are in blue.

**Response to Referee 2:**

*Philip Rupp and Thomas Birner*

*This is my second review of this manuscript. Thank you to the authors for their efforts in addressing my questions and concernts in the previous round of reviews. With the exception of a few small questions and some typos, I am satisfied by the present manuscipt and recommend publication. Line numbers below refer to the revised manuscript, not to the 'track-changes' version of the manuscipt, which do not agree; the latter seems to be a somewhat more recent version of the text and some of the typos identified below may have already been corrected.*

*l83: Can you be more explicit about how they disagree? This is not clear from the summaries given here.*

The for us most relevant difference in results is that Smy and Scott (2009) observe a decrease in linear growth rate of the life-cycle with increasing polar vortex strength (in terms of a stratospheric PV anomaly) while Wittman et al. (2007) report an increase in growth rate for increasing lower-stratospheric shear (and correspondingly increasing stratospheric wind speeds). We, on the other hand, do not observe any significant dependence of the growth rate on the stratospheric state in our experiments. The results by Wittman et al. and Smy and Scott can potentially be explained by a modification of the tropospheric state and/or near-tropopause conditions (as pointed out in Sections 3.2 and 5 of our revised manuscript, and also in the discussion of Smy and Scott). To further clarify these connections we changes the corresponding part as follows:

"They [Smy and Scott (2009)] found a decrease in growth rates and general wave activity, and a corresponding reduction in magnitude of the induced surface geopotential anomaly of the final state, with increasing strength of the stratospheric PV perturbation. Note that Wittman et al. (2007) reported an increase in growth rate with increasing stratospheric shear (and hence increasing stratospheric wind speed) for low synoptic wave numbers. However, Smy and Scott (2009) also note that some of their results (e.g., regarding sensitivity of growth rates) might be explained by a change in tropospheric horizontal shear due to the non-local effects of the stratospheric PV anomaly and a corresponding fundamental change in the nature of the life […]."

*Section 2: The authors' response to my previous comments suggest that their model is specified with an isobaric lower boundary; this is potentially of some quantitative relevance (e.g. Haynes and Shepherd 1989). If this is so it should be stated clearly in the model description.*

*Haynes, P. H. and T. G. Shepherd (1989) "The importance of surface pressure changes in the response of the atmosphere to zonally-symmetric thermal and mechanical forcing". Q. J. R. Meteorol. Soc. 115, pp. 1181-1208.*

We thank the referee for pointing out this subtlety of our model setup, which could indeed lead to confusion. The model we use to conduct the life-cycle experiments uses an isobaric lower boundary condition. However, since the model is formulated entirely in pressure coordinates

it does not include any actual definition of Earth's surface, which is the main reason that we need to approximate the surface response as the response at our lower-most model (pressure-)level, e.g., in terms of a geopotential height anomaly computed by meridional integration of the geostrophic zonal wind equation. We added the following sentence to the model description (Section 2) to clarify the precise lower boundary condition used in our model and comment on the approximation of the surface response in our diagnostics:

"Further note that the pressure coordinate formulation of the model used here lacks an explicit Earth's surface. When considering the surface response (e.g., in Section 3.3) we analyse the lowest pressure layer, thereby effectively approximating the actual surface response, which would require a modified physically consistent lower boundary condition (e.g., Haynes and Shepherd, 1989."

*l258, 281: "a simple way to quantify the eddy feedback processes" this claim is made in several places, but is not really expanded upon, and I am not sure exactly what the authors have in mind.*

Our claim is simply that the presented framework of idealised baroclinic life-cycles in a dry dynamical model can be used to study the tropospheric eddy feedback to stratospheric conditions, and hence quantify the corresponding processes that are potentially involved in a downward coupling of troposphere and stratosphere. One simple example is the quantitative study of the model energetics. To avoid potential confusion we slightly re-phrased the corresponding statements as follows:

"This jet shift is analogous to the NAM-like signature that has been observed after SSW events, and its appearance as response to stratospheric conditions in the framework of a dry dynamical model further indicates the fundamental importance of tropospheric synoptic-scale eddy feedback in causing the observed negative NAM-signal, as has previously been shown by other studies (e.g., Domeisen et al., 2013; Hitchcock and Simpson, 2014), and allows for a way to quantify these eddy feedback processes (e.g., in terms of EKE and MKE evolution) in a simple and idealised setting."

and

"It further provides a simple model framework to quantify the eddy feedback processes (e.g., in terms of EKE and MKE response) potentially involved in creating the corresponding jet shift signal."

We further corrected the following typos pointed out by the referee:
*l370: "stratospheric jet to substantially alter the" -> "stratospheric jet substantially altered the"*
*l373: This seems to be a note that was left in the text.*
*l448: extent*
*l487: life-cycle*